# Benchmarking Self-Supervised Video Representation Learning

## Abstract

Self-supervised learning is an effective way for label-free model pre-training, especially in the video domain where labeling is expensive. Existing self-supervised works in the video domain use varying experimental setups to demonstrate their effectiveness and comparison across approaches becomes challenging with no standard benchmark. In this work, we first provide a benchmark that enables a comparison of existing approaches on the same ground. Next, we study five different aspects of self-supervised learning important for videos; 1) dataset size, 2) complexity, 3) data distribution, 4) data noise, and, 5) feature analysis. To facilitate this study, we focus on six different methods along with six different network architectures and perform an extensive set of experiments on five different datasets with an evaluation of two different downstream tasks. We present several interesting insights from this study which span across different properties of pretraining and target datasets, pretext-tasks, and model architectures among others. Furthermore, we extend these findings to Video Foundation models (ViFMs). Finally, we put some of these insights to the real test and propose an approach that requires a limited amount of training data and outperforms existing state-of-the-art approaches which use 10x pretraining data. We believe this work will pave the way for researchers to a better understanding of self-supervised representation learning in videos.

## 1 Introduction

Deep learning models require a large amount of labeled data for their training. Obtaining annotations at large-scale needs a lot of effort and it becomes even more challenging as we shift from image to video domain. There are several interesting directions focusing on this issue such as domain adaptation (74), knowledge distillation (20), semi-supervised learning (77), self-supervision (31) and weakly-supervised learning (56), which attempts to rely on the knowledge learned from existing source datasets and transfer to new target datasets with minimal labels. Among these approaches, self-supervised learning use pretext task as supervisory signal and does not require any labels on source datasets which makes it more favorable.

In recent years, we have seen great progress in self-supervised learning (SSL) in video domain (75; 32; 78; 69; 49; 10). More recently, the focus is more towards context-based learning which involves modifying input data such that to derive a classification (73; 13; 75; 32), reconstruction (78; 10) or generative (67; 58; 24; 63; 46) signal which can be used as a learning objective. The main focus of these works is designing a pretext task that is computationally inexpensive and which provides a strong supervisory signal such that the model learns meaningful *spatio-temporal* features.

Submitted to the 38th Conference on Neural Information Processing Systems (NeurIPS 2024) Track on Datasets and Benchmarks. Do not distribute.

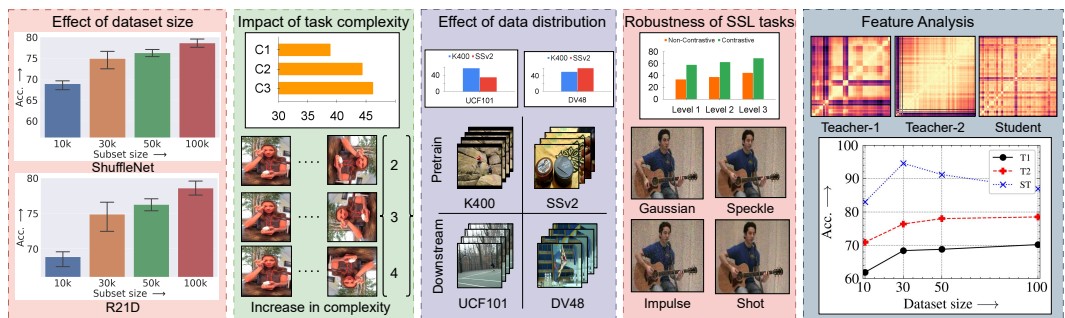

Figure 1: **Overview of proposed benchmark.** We study five different aspects in this benchmark study. Starting from left, 1) we show the analysis of *effect of dataset size vs training time*. As the dataset size increases, variation in performance decreases even with longer training time, 2) We show the effect of *task complexity* (C1, C2, C3 - Different complexities). Bottom figure shows use case of how complexity increases for the RotNet task, and, top figure shows how the performance varies for the R21D network, 3) With different *data distribution shifts*, the third sub-figure shows the impact of *target* data distribution on the *source* data, 4) We look into another data distribution shift due to introduction of noise. We see how *non-contrastive* tasks are more robust than *contrastive* ones even with increasing levels of severity of noise. The bottom part shows an example for each type of noise. Clips are provided in supplementary, and, 5) Finally, we further analyze whether the features learn *orthogonal* information. In this sub-figure, we show that using different architectures as teachers can substantially improve performance even in a low-data regime.

Despite this great progress, it is non-trivial to compare these approaches against each other due to a lack of standard protocols. These methods are evaluated under different conditions and there is no standard benchmark to evaluate the fair effectiveness of these methods. A recent study (62) attempts to take a step towards this direction, but it is mainly focused on downstream learning, without exploring the self-supervision aspect which is one of the main goals in our study. In this work, we present a benchmark where important self-supervised pre-training parameters are kept consistent across methods for a fair comparison. With the help of this benchmark, we study several critical aspects which are important for self-supervised learning; *1) effect of pretraining dataset size, 2) task complexity, 3) generalization under distribution shift, 4) robustness against data noise, 5) properties of learned features.* Fig. 1 provides an overview.

The proposed benchmark includes a large-scale assessment of context-based representative self-supervised methods for video representation learning. We analyze two different factors: 1) *learning objective* which includes *contrastive* vs *non-contrastive*, and 2) *data transformation* that comprises three categories namely, *spatial*, *temporal*, and *spatio-temporal*. We study six different pretext tasks with six different models and perform our experiments on five different action recognition datasets and evaluate these approaches on two different downstream tasks, action recognition, and video retrieval. Furthermore, we extend the study to recently developed video foundation models.

We observe some interesting insights in this benchmark; 1) Contrastive tasks are fast learners but are less robust against data noise, 2) there is no direct relation that increase in pretext task complexity leads to better understanding of spatio-temporal representation learning, 3) *temporal* based pretext tasks are more difficult to solve than *spatial* and *spatio-temporal*, 4) spatio-temporal task can solve the pretext task independent of data distribution shifts, and finally, 5) we empirically show that these pretext tasks learn complementary features across factors such as model architecture, dataset distributions, dataset size, and pretext task. Our contributions are threefold:

- We present a benchmark for self-supervised video representation learning to compare different pretext tasks under a similar experimental setup.

- We perform extensive analysis on 5 important factors for self-supervised learning in videos; 1) dataset size, 2) task complexity, 3) distribution shift, 4) data noise, and, 5) feature analysis.

- Finally, we put some of our insights from this study to test and propose a simple approach that outperforms existing state-of-the-art methods on video action recognition with a limited

amount of pretraining data. Additionally, based on our findings, we put down a set-up recipe for future self-supervised learning algorithms to build upon.

## 2 Related Work

**Self-supervised learning** There are several works in the domain of self-supervised learning for video representation learning (31; 55). These approaches can be grouped into two main categories on the basis of pretext task: 1) context-based (34; 71; 3; 19; 73; 61; 76; 13; 30; 69; 49; 10; 16; 23; 50), and 2) cross-modal (48; 53; 1). Cross-modal approaches use multiple modalities such as audio, video, optical flow, and camera positions, and rely on consistencies across these modalities. Context-based learning exploits data transformations to derive supervisory signals for training the model. Context-based pretraining tasks have evolved a lot in the past few years. Our work explores the domain of how much variation in learned representations under different transformations. In contrast to other approaches, context-based approaches exploit the spatial and temporal information independently by several transformations (43; 19; 75; 7; 73; 49; 69). Recent works have started to transform the spatial and temporal domain together (34; 42; 61; 78; 10). Incorporating multiple modalities improves performance, but, it's not available for all datasets, especially large-scale datasets. In this work, we restrict our focus to single-modality (RGB) approaches.

**Self-supervised benchmarking** There are some prior efforts focusing on benchmarking self-supervised learning in the image domain. In (21), the authors provide a detailed analysis of image-based self-supervised learning approaches and study how dataset size scaling affects the learned representations. Similarly in (35), the authors analyze how different model architectures play a role in visual self-supervised learning. In both these works, the authors did not focus on the importance of various pretext tasks themselves but only showed how certain pretext tasks can be improved. Therefore, their main focus was on downstream tasks rather than pretext learning. We, on the other hand, study different pretext tasks and analyze how various aspects affect feature learning. Moreover, these works are focused on the image domain, whereas we focus on the video domain. In recent work, (18), a study was performed to better understand unsupervised learning in the video domain. It explored the use of several pre-text tasks from the image domain and applied them to videos. We are not merely focusing on down-stream tasks and our attention is on the self-supervised aspect which includes factors such as data subset size, task complexity, dataset distribution, and noise robustness.

## 3 Self-Supervised Configurations

We first describe the pretext tasks used in our study along with their categorization followed by details of this benchmark including network architectures, datasets, downstream tasks and evaluations.

### 3.1 Tasks categorization

We analyze two different aspects of video pretext tasks: 1) transformations applied to data, and 2) learning objectives. Data transformations include, *spatial-based (S)*, *temporal-based (T)* and *spatio-temporal (ST)*. *Spatial* transformations include reshuffling of spatial patches, temporal consistent data augmentation, or rotation of images/patches. *Temporal* tasks involve permutation classification of frames/clip, order verification, clips sampling at different paces, or, contrastive learning from temporal triplets. *Spatio-temporal* tasks include those in which we modify both of these parameters simultaneously. This includes dilated sampling and simultaneous frame reconstruction, shuffling spatial and temporal domains, or, speed prediction, and contrastive visual features. Learning objectives can be either *contrastive* (11) or *non-contrastive* such as (78).

Following this categorization, we select at least two representative pretext tasks from each *transformation* category, one *contrastive* and one *non-contrastive*. We study the following pretext tasks: RotNet (Rot) (32), Video Clip Order Prediction (VCOP) (75), Playback Rate Prediction (PRP) (78), Spatiotemporal Contrastive Video Representation Learning (CVRL) (49), Temporal Discriminative

Learning (TDL) (69) and Relative Speed Perception network (RSPNet) (10). The description of tasks are provided in the supplementary (Section C).

## 3.2 Benchmark details

This section standardizes the conditions used by our benchmark to compare different pretext tasks. Further explanation for using these conditions are outlined in the supplementary.

*Datasets:* We experiment with two different dataset types, 1) where appearance is more important, and 2) where time is more important. For appearance based, we use Kinetics-400 (33), UCF101 (57), and HMDB51 (38), where appearance is more important (recognize activity with a single frame) than temporal aspect, and for temporal aspect, we use Something Something-V2 (22) and Diving48 (39), where temporal information plays a significant role (require few frames to recognize activity). More details are in the supplementary.

*Spatio-temporal architectures:* We consider three different network capacities, 1) small-capacity, 2) medium-capacity, and large-capacity. For small capacity networks, we use ShuffleNet V1 2.0X (79), whereas for medium capacity we focus on R(2+1)D (65) (R21D). We do not include large capacity networks in our main benchmark in the interest of computational efficiency; additional results for such a model, VideoSwin (41) is shown in the supplementary.

*Downstream tasks:* We show results and analysis on two different downstream tasks - *action recognition* and *clip retrieval*. These two tasks are the most prominent in the field of self-supervised learning in videos. Full finetuning is performed as opposed to linear probing to adapt models.

*Evaluation and Analysis:* We use top-1 accuracy for action recognition and top-K for Clip retrieval. For robustness performance, we calculate the relative robustness score $(R_s)$ using original accuracy on clean test set $(A_c)$ and perturbed accuracy on noisy test set $(A_p)$ as $R_s = \frac{A_c - A_p}{A_c}$. Centered Kernel alignment (CKA) (44) maps illustrates model behaviours. More details in supplementary.

## 4 Benchmark Analysis

In this section, we perform analysis across the following five aspects:

*Effect of pretraining dataset size:* In self-supervised learning, a natural question to ask is whether dataset size plays any role in the performance of downstream tasks. It is important to study if the increase in the size of the pretraining dataset will proportionally reciprocate in performance improvement. Also, a general trend is to train models for a very long duration at the pre-training stage. We investigate if the longer duration actually impacts the gain in performance. We look across different stages of training for multiple architectures and across different pretext tasks.

*Impact of task complexity:* Some of the existing works show that increasing complexity leads to better representation learning, and if the complexity is decreased, the network will optimize to suboptimal solutions. We analyze this aspect in more detail with several tasks and different models.

*Effect of data distribution:* Existing self-supervised methods perform evaluations on K400 and UCF101 datasets. Both these datasets fall into the same visual category with heavy appearance bias. However, we divert our attention towards datasets where the temporal dimension plays an important role such as SSv2 and Diving48.

*Robustness of SSL tasks:* We study the robustness qualities of SSL methods against data noise (26). We analyze which factors play a key role in robustness of these methods against such domain shifts.

*Feature analysis:* Finally, we look into feature space and analyze whether the learned representations are complementary in nature when models are trained under different protocols.

### 4.1 Effect of dataset-size

We first analyze the effects of pre-training data size variation. The network trains on four subsets of the K400: 10k, 30k, 50k, and 100k. The number of videos per class is the same. The smaller pre-training dataset is a subset of the bigger pre-training dataset size (i.e. $10k \subset 30k$ and so on). We look into three aspects regarding *dependence on pre-train subset size:* a) behavior of different pretext tasks with the increase in pre-train dataset subset, b) performance across the different capacity of backbones, and, c) the effect of training time across different pretext tasks.

Table 1: Evaluation of different pretext tasks on **different subset size** on R21D network (%).

| | Non-Contrastive | | | Contrastive | | |
|---|---|---|---|---|---|---|
| Subset | Rot | VCOP | PRP | CVRL | TDL | RSPNet |
| 10k | 37.6 | 46.3 | 17.5 | 55.9 | 31.1 | 70.9 |
| 30k | 36.2 | 50.4 | 42.7 | 56.9 | 30.9 | 76.4 |
| 50k | 41.2 | 51.5 | 46.2 | 61.2 | 30.2 | 78.0 |

Table 2: **Performance at different stages** of training for all pretext tasks on R21D (50k)(%).

| | Non-Contrastive | | | Contrastive | | |
|---|---|---|---|---|---|---|
| Epochs | Rot | VCOP | PRP | CVRL | TDL | RSPNet |
| 50 | 35.4 | 52.2 | 24.1 | 55.7 | 32.1 | 75.0 |
| 100 | 37.3 | 52.3 | 34.8 | 58.5 | 31.3 | 76.1 |
| 150 | 40.7 | 51.3 | 46.7 | 60.2 | 31.5 | 76.5 |
| 200 | 40.9 | 52.8 | 45.0 | 60.5 | 30.2 | 77.4 |

Table 3: **Complexity Variation.** TC: Task complexity. Results are shown on UCF101 with ShuffleNet/R21D backbone.

| TC ↓ | S | T | ST |
|---|---|---|---|
| C1 | 20.1/48.3 | 41.6/**56.8** | **24.2**/38.9 |
| C2 | **20.2/58.3** | **41.8**/54.8 | 18.1/44.4 |
| C3 | 16.6/41.2 | 40.6/55.6 | 21.9/**46.2** |

**Observations:** From Table 1, we observe that apart from TDL each pretext task performance improves with an increase in subset size. If we look into specific pretext task transformation category (Table 1), the most gain with an increase in data is for *spatio-temporal* tasks ( 13%), whereas the least gain is for *temporal* pretext tasks ( 3%). Analyzing the effect of the duration of training across different tasks, in Table 2, the performance gain is minimal (<1.5%) after training for more than 100 epochs. Comparing contrastive and non-contrastive approaches, the gain in contrastive-based approaches is on average 1% compared to 5% for non-contrastive tasks beyond *100 epochs* of training.

**Inference:** (i) *Spatio-temporal pretext tasks improve most with increment in dataset size and are most dependent on it than others since it involves transformation along both axes: appearance (spatial) and motion (temporal).* (ii) *Contrastive tasks are fast learners against non-contrastive and reach their potential in a relatively shorter duration.*

### 4.2 Impact of change in task complexity

Next, we study the effect of task complexity. In this aspect, we analyze only non-contrastive tasks as it is non-trivial to define task complexity for contrastive-based approaches. We analyze three different complexities (C1, C2, C3) for each task. The variation in complexity for each task is briefly discussed as follows: a) *RotNet*: vary the number of rotations between 2 to 4, b) *VCOP*: increase the number of shuffle clips from 3 to 5, and, c) *PRP*: modify the dilation sampling rates from 2 to 4 classes. We investigate the following aspects here: a) does an increase in complexity means better spatio-temporal features learned at the pre-training stage? b) does the capacity of architecture plays any role?

**Observations:** From Table 3, comparing across rows we observe ShuffleNet performance doesn't improve much or degrade significantly if the complexity of the task is increased. CKA maps show the structure transforms from staggering grids to a multi-block pattern indicating saturation with an increase in complexity. In between different categories of transformation, performance improves with complexity for the bigger model in the case of the *spatio-temporal* task. Between ShuffleNet and R21D, R21D gives staggering grids against dark block patterns for ShuffleNet which shows the model can still learn better features. CKA maps are provided in the supplementary.

**Inference:** (i) *Increase in pretext task complexity doesn't always reciprocate to better spatio-temporal feature learning. It is dependent on the pretext task and also the model capacity.* (ii) *If higher complexity improves features learning, the model should also have the capacity, otherwise the task will be too difficult for the model to learn meaningful representations.*

### 4.3 Effect of dataset distribution

Shifting our focus to datasets that have more hidden cues in the temporal aspect, we add pre-training on SSv2 and finetuning on Diving48 to our experiments. We answer the following questions in this section; a) does the categorization of pretext-task matter on *source (pre-training)* and *target (downstream)* datasets? b) what is the impact of *source* dataset when the pretext task focuses only on a single task either *spatial* or *temporal*?

**Observations:** Looking into Figure 2, we observe that *spatio-temporal* pretext task outperforms other pretext tasks on both *target* (downstream) datasets UCF101 and DV48 by a margin of 15-40% and 10-13% respectively whether the *source* datasets is K400 or SSv2. Comparing, spatial and temporal-based pretext tasks, we see that they are *majorly* dependent on *source* datasets. Looking at Figure 2, performance is better on both *target* datasets if *source* dataset has the same underlying

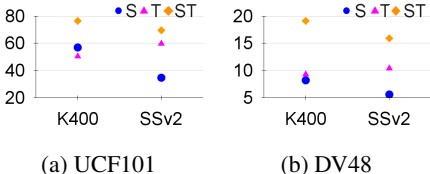

(a) UCF101      (b) DV48

Figure 2: **Effect of different dataset distributions:** Here, S, T, and ST mean spatial(CVRL), temporal(VCOP), and, spatio-temporal(RSPNet) respectively. X-axis shows *source* dataset and Y-axis shows Top-1 accuracy.

|  | Non-Contrastive | | | Contrastive | | | |
|---|---|---|---|---|---|---|---|
|  | Rot | VCOP | PRP | CVRL | TDL | RSP | Avg. |
| R21D | 10.7 | 19.0 | 70.1 | 78.4 | 26.7 | 68.8 | 45.6 |
| Shuffle | 28.3 | 28.4 | 22.8 | 51.9 | 43.5 | 28.6 | 33.9 |

Table 4: **Robustness analysis** on the relative decrease in % performance across different pretext tasks on noisy UCF101 dataset. The performance is averaged over 4 noises.

properties as the pre-text task is trying to learn. Furthermore, the spatial task is more dependent on the *source* dataset, since the relative drop on both UCF101 and DV48 for CVRL is significant (40% and 30% respectively) when the source dataset is SSv2 against K400. However, in the case of the temporal task, the drop is 15% and 10% respectively when the source dataset is K400 against SSv2.

**Inference:** (i) *Spatio-temporal pretext task learns better features independent of source and target data distribution.* (ii) *Spatial and temporal pre-text tasks are better learners when source data distribution belongs to spatial and temporal respectively.* (iii) *Temporal pretext task prevails when target data is temporal, whereas, spatial is dependent on source data distribution.*

### 4.4 Robustness of SSL tasks

Similar to OOD datasets, introducing noise also shifts the distribution of datasets. We evaluate models on different types of noises introduced in (54) with different severity levels on the UCF101 test dataset. Specifically, we probe into four different types of appearance-based noises: Gaussian, Shot, Impulse, and Speckle (26). Here we look into the following aspects: a) how robust different categorizations of pretext tasks are? b) is the network's architecture dependent on the noise in the dataset? In the main paper, we only discuss one severity level and have provided a detailed analysis of multiple severity levels in the supplementary.

**Observations:** From Table 4, we observe that the relative drop in performance for contrastive tasks is more than non-contrastive tasks for both R21D and ShuffleNet backbone. The most and least robust models are RotNet-R21D and PRP-R21D with 10.7% and 70.1% relative decrease.

**Inference:** *Contrastive approaches are less robust to noise as compared with non-contrastive.*

### 4.5 Feature analysis

We further analyze the learned features by these pretext tasks under different configurations. We specifically focus on understanding the complementary nature of these features. We employ knowledge distillation (15) as a tool to study this aspect. It is based on the idea that distilled knowledge from the ensemble of teacher networks makes the student model stronger. The loss function for multi-teacher knowledge distillation is: $\mathcal{L}_{KD} = \mathcal{L}_{CE} + \mathcal{L}_{KL_1} + \mathcal{L}_{KL_2} + ... + \mathcal{L}_{KL_n}$, where $\mathcal{L}_{CE}$ is the cross-entropy loss for hard labels and $\mathcal{L}_{KL_n}$ is the KL-Divergence loss for soft labels from teacher $n$. We use our benchmark models as teachers in different combinations to analyze whether a student learns orthogonal information on four different axes: 1) different architectures as the teacher within a *dataset size*, 2) teachers with different complexities in a pretext task, 3) models from multiple *source* datasets, and, 4) same architecture as teachers from multiple pretext tasks. Figure 3 summarizes the *observations* for each aspect. More details are in supplementary.

**Observations:** Although teacher network performance improves with subset, gain in complementary information reduces beyond 30k (Figs. 4(a) & 4(b)). However, distillation does help in the reduction of training time with a significant improvement in performance which is evident from Fig. 3(a). Independent of the pretext tasks category smaller architecture learns complementary information and outperforms the teacher whereas bigger architecture it's task-dependent. Irrespective of task category whether transformation-based or contrastive, each task learns corresponding features

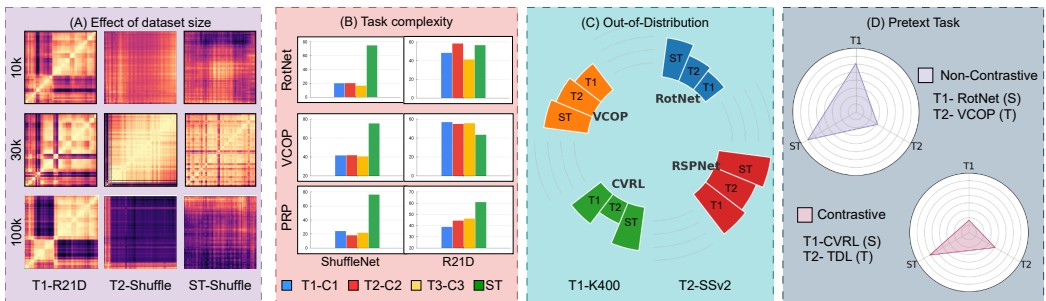

Figure 3: **Feature analysis overview.** This figure shows how KD as a tool is beneficial across multiple scenarios. Brief details for each setup (Left to right): (A) *Effect of dataset size:* Teachers (T1 and T2) are different architectures for a single subset. Student model (ST-Shuffle) CKA maps shows it learns complementary information especially for 30k. (B) *Task Complexity:* Teachers are multiple complexities across the same task. (C1, C2, C3 - different complexities as teachers.) We observe in most of the scenarios, Student (ST) networks outperforms all teacher models which proves learning of orthogonal information from multiple teachers. (C) *Out-of-Distribution:* Models from different *source* datasets are teachers. Student model (ST) outperforms both teachers trained on two different datasets. (D) *Pretext Tasks:* Spatial and temporal task networks are teachers, and, student model (ST) learnt from two different categories of pretext tasks - spatial and temporal incorporate knowledge from both and outperforms both of the teachers for both contrastive and non-contrastive.

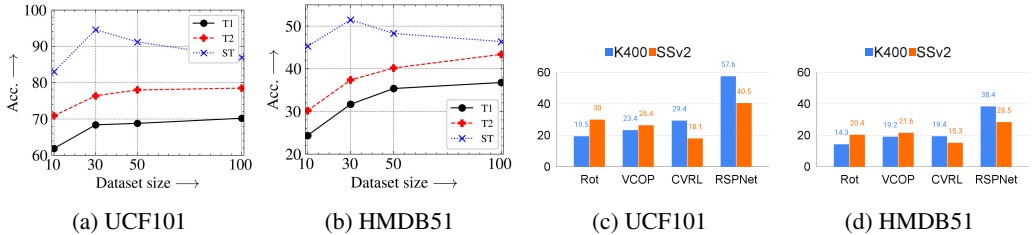

| (a) UCF101 | (b) HMDB51 | (c) UCF101 | (d) HMDB51 |

Figure 4: **Knowledge distillation** using teachers trained on multiple subset sizes on RSPNet. Student: ShuffleNet a) UCF101 and b) HMDB51. Here T1 is Teacher-1 (shufflenet) and T2 is teacher-2 (R21D). **Top@5 Clip Retrieval** - R21D on c) UCF101 and d) HMDB51, pre-trained on K400 and SSv2 - 30k subset.

from both source datasets and outperforms the teacher. Student network outperforms standalone spatio-temporal network performance in both contrastive and non-contrastive domains.

**Inference:** (i) *Knowledge can be distilled from different architectures for a given subset size (Fig. 3 (a))*, (ii) *Knowledge from different source datasets brings in complementary information (Fig. 3 (c))*, and (iii) *Orthogonal features are learned across different categories of pretext tasks (Fig. 3 (d))*.

## 5   Lessons Learned

With all the analysis along studied axes, we learned a few lessons in-between these axes such as: (i) Contrastive tasks are fast learners but are also most susceptible to noise. (ii) An increase in dataset size or complexity does not help smaller models in learning better spatio-temporal features but these features are more robust to noise. (iii) Temporal tasks are relatively more difficult to learn since looking at the correlation between time of training, increase in dataset size, and complexity, the performance gain is minimal in each of this axis. It means this category of tasks is actually difficult to solve. (iv) Spatio-temporal pretext tasks improve with the increase in complexity and dataset size (if the model permits), and their behavior to learn better spatio-temporal features is independent of data distribution. Using these lessons, we further do more analysis in feature space. From there, we observe within an axis of comparison how models learn orthogonal information. Based on those observations, we analyze if we can push the performance for downstream tasks. We look into two downstream tasks: action classification and clip retrieval.

Table 5: **Comparison with previous approaches** pre-trained on K400. Ours ( * best performing) is RSPNet pretrained on 30k subset of K400. † - Different pre-training data. (%)

| Approach | Venue | NxW/H | Backbone | Pre-training | UCF101 | HMDB51 |
|---|---|---|---|---|---|---|
| **Generative** | | | | | | |
| VIMPAC (60) | - | 10x256 | ViT-L | HTM | 92.7 | 65.9 |
| VideoMAE (63) | NeurIPS'22 | 16x224 | ViT-B | K400 | 91.3 | 62.6 |
| MME (59) | CVPR'23 | 16x224 | ViT-B | K400 | 96.5 | 78.0 |
| MVD (70) | CVPR'23 | 16x224 | ViT-B | IN1K+K400 | 97.0 | 76.4 |
| EVEREST (28) | - | 16x224 | ViT-B | - | 93.4 | 68.1 |
| SCE (14) | WACV'23 | 16x224 | ResNet3D-50 | K400 | 95.3 | 74.7 |
| **Context** | | | | | | |
| PacePred (73) | ECCV'20 | 16x112 | R21D-18 | K400 | 77.1 | 36.6 |
| TempTrans (30) | ECCV'20 | 16x112 | R3D-18 | K400 | 79.3 | 49.8 |
| STS (68) | TPAMI-21 | 16x112 | R21D-18 | K400 | 77.8 | 40.5 |
| VideoMoCo (46) | CVPR'21 | 16x112 | R21D-18 | K400 | 78.7 | 49.2 |
| RSPNet (10) | AAAI'21 | 16x112 | R21D-18 | K400 | 81.1 | 44.6 |
| TaCo (6) | - | 16x224 | R21D-18 | K400 | 81.8 | 46.0 |
| TCLR(13) | CVIU'22 | 16x112 | R21D-18 | K400 | 88.2 | 60.0 |
| CVRL (49) | CVPR'21 | 32x224 | R21D-18 | K400 | 92.9 | 67.9 |
| TransRank (16) | CVPR'22 | 16x112 | R21D-18 | K200 | 87.8 | 60.1 |
| **Multi-Modal** | | | | | | |
| AVTS (37) | NeurIPS'18 | 25x224 | I3D | K400 | 83.7 | 53.0 |
| GDT (47) † | - | 32x112 | R21D | IG65M | 95.2 | 72.8 |
| XDC (4) | NeurIPS'20 | 32x224 | R21D | K400 | 84.2 | 47.1 |
| Ours * | - | 16x112 | R21D-18 | K400-30k | 97.3 | 51.5 |

**Clip retrieval** For this downstream task, we generate feature vectors using pretrained weights. The nearest neighbor is found by measuring the cosine distance between test and train feature vectors. We show analysis on UCF101 and HMDB51, with different source data distributions, K400 and SSv2. *Observations:* Spatio-temporal task still outperform other categories independent of *source* data distribution similar to what we observe earlier. Contrastive learns better *appearance* features during the pre-training stage given both downstream datasets are *appearance* based. Temporal tasks have almost similar performance pre-trained on either of the *source* datasets, which shows even with an appearance-based dataset as a pre-train dataset, the task is not focusing much on spatial features.

**Action Classification** For this task, the model is finetuned end-to-end on downstream datasets, on UCF101 and HMDB51. In Table 5, we obtain our best performing model via knowledge distillation discussed in previous section and we show our model outperforms previous state-of-the-art approaches. *Observations:* With only 30k videos compared to 200k+ videos used by other pretext tasks, we show that our model outperforms by a good margin on UCF101 against single and multi-modal approaches. We got competitive results on HMDB51 with a score of 51.5%.

## 5.1 Surprising Findings

We have multiple inferences from different axes of analysis. However, to club a few which are new and helpful for video self-supervised community, we list down those here:

*Dataset size and Training time Dependency:* Against the conventional belief that a lot of training data is a *must* to achieve the best performance, we demonstrate that beyond a certain amount of training data, additional data provides diminishing returns for SSL in terms of performance improvement. This finding has significant implications, as it allows for a substantial reduction in the training data and there is almost a 10x reduction in training time which is particularly advantageous in computationally demanding video processing tasks. Furthermore, we show how KD as a tool, outperforms the original approach (100% data) using almost 90% less data further optimizing resource utilization by 80%.

*Robustness to real-world noise:* To our surprise, contrastive tasks are more susceptible to noise than non-contrastive. A smaller network tends to be more robust in some scenarios than a bigger network. We believe these findings are *novel and not known* to the community as there is no existing study exploring these aspects and are helpful where robustness is necessary for real-world deployment.

Table 6: **Analysis on ViFMs**. Zero-shot classification accuracy on UCF-101. I:Image, V: Video.

| | ViFM | Type. | Pretraining Data | Frames x Rate | Accuracy |
|---|---|---|---|---|---|
| ViFi-CLIP (51) | | I | K-400 | 32 x 2 | 77.3 |
| X-CLIP (45) | | I | K-400 | 8 x 8 | 71.6 |
| EZ-CLIP (2) | | I | K-400 | 8 x 8 | 70.5 |
| ViCLIP (72) | | V | InternVid-10M | 8 x 8 | 75.5 |
| LanguageBind (80) | | V | VIDAL-10M | 8 x 8 | 69.9 |

Table 7: **Knowledge Distillation** between different ViFM pairs as teachers, and R21D as the student.

| ViFM | X-CLIP | ViFi-CLIP | EZ-CLIP | ViCLIP | LanguageBind |
|---|---|---|---|---|---|
| X-CLIP | X | 83.2 | 88.7 | 88.2 | 87.6 |
| ViFi-CLIP | X | X | 88.0 | 86.6 | 86.6 |
| EZ-CLIP | X | X | X | 85.0 | 86.9 |
| ViCLIP | X | X | X | X | 85.4 |
| LanguageBind | X | X | X | X | X |

*Complementary knowledge:* Improvement in performance with KD from different data distributions and categories of tasks brings out a recipe for a new SSL task. This involves utilizing a multi-teacher multi-student setup, where each teacher specializes in spatial and temporal tasks and is trained on a mixture of data sources. Our analysis indicates this would provide a strong learning scenario.

## 5.2 Recommendations

Looking into several factors, here we provide a few recommendations to set up the recipe for SSL: 1) *Training speed:* If training time is a concern, contrastive tasks can help in reducing the pretraining time, but they could be less robust against data noise. 2) *Data distribution:* It is always better to use a spatio-temporal pretext task irrespective of the data distribution. However, if that is not an option, the pretext task should always be aligned with the nature of the pretraining dataset. 3) *Model capacity:* If model capacity is limited, there is no benefit of increasing pretraining dataset size and using complex pretext tasks. 4) *Robustness:* If best performance is the goal, we should use a non-contrastive as opposed to a contrastive pretext task. 5) *Performance:* Pretext tasks learn complementary features across model architectures, pretraining datasets, pretext tasks, and tasks complexity, therefore, this complementary knowledge can be distilled to obtain strong spatio-temporal features.

## 5.3 Extension of findings to Video Foundation Models (ViFMs)

In this section, we extend the study to ViFMs (Tables 6 and 7). We select both image-based (2; 45; 51) which are image foundation models extended to videos and video-based (80; 72) which are trained from scratch on videos. ViFMs are all trained with contrastive pretraining objective. More details about architectures are in supplementary.

*Dataset size:* An increase in dataset size or complexity does not help smaller models in learning better spatio-temporal features (Table 6). ViCLIP and LanguageBind, despite using a significantly larger pretraining dataset, performs worse than models pretrained on the smaller Kinetics-400 dataset; A simple increase in the number of frames is outperforms models trained on larger datasets.

*Complementary knowledge:* Improvement in performance in the case of KD from different ViFMs brings out a recipe for training a new foundational model. This involves utilizing a multi-teacher multi-student setup, where each teacher is a ViFM pretrained differently in terms of data sources, multi-stage pretraining, and pretraining objective. Our analysis (Table 7) indicates this would provide a powerful learning scenario.

## 6 Conclusion

In this study, we explore different parameters for self-supervised learning in the video domain. We set a benchmark which provides an intuitive task categorization and enables a better comparison of different pretext tasks. Such an analysis has never been explored for video understanding to the best of our knowledge. We presented several interesting insights which will open up new directions for the research community. We also demonstrate the usefulness of some of these insights where we obtain state-of-the-art performance on video action recognition using merely a 10% pretraining dataset when compared with existing methods. We believe this benchmark study will help the research community better understand self-supervised learning in the video domain.

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
