# OpenReview forum: "Benchmarking Self-Supervised Video Representation Learning"
_NeurIPS.cc/2024/Datasets_and_Benchmarks_Track — Submitted to NeurIPS 2024 Track Datasets and Benchmarks_

### Official Review · Reviewer_Kssv · 2024-07-20
**Benchmarking some SSL methods for videos. Overall, the experiments and analysis appear incomplete.**

**Rating:** 6
**Confidence:** 5
**Correctness:** The design is not appropriate, please…

**Review:**

1. The numbers are acceptable: 6 methods, 5 tasks, 5 benchmark datasets. However, my main concern is the completeness of the primary task, which is SSL video model evaluation. This is a comprehensive field and thus raises many questions about the design choices, their completeness, and consequently the soundness of the generalizations or conclusions. These aspects are largely missing.
2. I think the authors should spend more time on the literature section. Avoid clubbing references together, and present a tabular chronological development of the field. This will help address many issues:
    - (a) Clearly define the baselines
    - (b) Clearly define the categories of methods
    - (c) Provide a more understandable rationale behind the choice of methods, ablations, and datasets
    - (d) Ground the work more appropriately in the literature, making it useful to the community
3. Furthermore, a table should be added summarizing the latest studies on the same topic and their contributions. Based on this, the identified gaps should be reasonably addressed.
4. The authors also need to position their observations with respect to other studies on SSL and their findings.
5. The term "feature analysis" and its context are not well explained, even though it is frequently used from beginning.
6. The choice of methods is questionable, leading to several basic limitations:
    - Due to the choice of models, the studied effects are limited to those models only. How can these findings be generalized to various other classes of models?
    - A discussion should be added justifying the selection of these methods and tasks while ignoring others, to establish the representative completeness of these tasks.
8. Ablations are limited to noise only, while many other perturbations exist. This needs to be justified.
9. Citations to previous studies on the same topic are missing. Many studies have addressed SSL for videos.
10. The current "context-based and cross-modal" classification scheme is limited, as context-based and cross-modal are not mutually exclusive. Further classifications like generative vs. discriminative or equivariance vs. invariant-based should also be covered. Additionally, hybrid approaches that merge context and modalities should be included. The field is vast, and many design choices need to be explained. It is advisable to cover a niche thoroughly, especially if it overlaps with futuristic trends, as this will be more helpful for the community.
11. Terms like Rot, VCOP, PRP, TDL, and metrics used in Tables 1 and 2, as well as abbreviations like S, T, and ST, should be defined close to their first use (preferably in the captions). The presentation currently lacks clarity.
12. The dataset used for the experiments should be mentioned in each table, e.g., Table 1, 2, etc.
13. The use of methods across different experiments is not uniform; some are missing.
14. The terms in Figure 1 should at least be expanded (if not explained) in the captions. It is cumbersome to search for them throughout the paper.
15. In the abstract, it is claimed that "...we focus on six different methods along with six different network architectures...", but only 2 methods, ShuffleNet and R2+1D, are used. Moreover, both are from 2018, which is quite old.
16. There is a contradiction: in the abstract, it is stated, "...Furthermore, we extend these findings to Video Foundation models (ViFMs)...", but elsewhere it is mentioned, "...We do not include large capacity networks in our main benchmark in the interest of computational efficiency...".
17. Grammar should be corrected, such as "... increase in the number of frames is outperforms models...".

**Strengths:**

I appreciate that the authors have organized this strategy considering different methods, datasets, and aspects of datasets and methods.

**Additional Feedback:**

Please see main comments.

**Clarity:**

Paper lacks clarity in several places. Main terms are not defined, several acronyms are not defined. Table captions are incomplete. Writing is more verbose and lacks simplicity and precision.

**Documentation:**

Not a dataset paper.

**Ethics:**

No concern.

**Limitations:**

Overall, the experiments and analysis appear incomplete. Literature is not well studied. Several choices are not well reasoned.
As the authors propose a new model, the paper is more suitable for non-dataset/benchmark tracks.

**Opportunities For Improvement:**

The current "context-based and cross-modal" classification scheme is limited, as context-based and cross-modal are not mutually exclusive. Further classifications like generative vs. discriminative or equivariance vs. invariant-based should also be covered. Additionally, hybrid approaches that merge context and modalities should be included. The field is vast, and many design choices need to be explained. It is advisable to cover a niche thoroughly, especially if it overlaps with futuristic trends, as this will be more helpful for the community.
Further there are many opportunities to improve presentation and various design choices and reorganization of the whole study. Please refer to main comments.

**Relation To Prior Work:**

Significantly missing.

**Summary And Contributions:**

This paper proposes to benchmark self-supervised learning (SSL) methods for videos. It uses various datasets and methods, testing them on two downstream tasks, and evaluates their sensitivity to dataset size, noise, task complexity, data distribution, and features.
Main contribution:
- Considering different aspects of SSL model training.

---

> ### Author Rebuttal · Authors · 2024-08-16
>
> We sincerely thank the reviewer for the valuable feedback and analysis of the paper. We have addressed the questions and concerns raised by the reviewer.
>
> **W1. Query about design choices, their completeness, and consequently the soundness of the generalizations or conclusions. These aspects are largely missing.**
>
> - Query about design choices: Our study focuses on context-based SSL approaches. We looked over 40+ SSL approaches context-based and then came up with two major taxonomies for design choices, namely: 1) Data transformation during the pre-training stage: Spatial, Temporal, and Spatio-temporal, and, 2) Contrastive and Non-contrastive approaches. We selected architectures that have been used by most of the context-based approaches. The SSL models are evaluated on five benchmark datasets for action recognition. There’s always a scope for the inclusion of more architecture and pretext tasks, but that would make the study infeasible. We cover all the variations in terms of the taxonomy we base our study on.
>
> - Soundness of conclusions: Both our quantitative and qualitative analysis show multiple practical insights that could be very useful for self-supervised domains. Reduction in train time, data requirement of architectures, and effect of testing models in out-of-distribution settings. These all correspond to real-world settings plus the existing work going on in the self-supervised domain. From our insights, we got the best results with limited resources.
>
> **W.2 Present a tabular chronological development of the field of**
> (a) Clearly define the baselines
> (b) Clearly define the categories of methods
> (c) Provide a more understandable rationale behind the choice of methods, ablations, and datasets
> (d) Ground the work more appropriately in the literature, making it useful to the community
>
> - Baselines: We ran a preliminary analysis on the K400 50k subset for fair comparison evaluation on both shufflenet and R21D. The table below shows the results:
>
> | Network    | Rot| VCOP | PRP | CVRL | TDL | RSP |
> |---------------|-------|--------|------|------|------|------|
> | Shuffle    | 16.6 | 40.8 | 21.9 | 62.3 | 12.4 | 68.8 |
> | R21D       | 41.2 | 51.5 | 46.2 | 61.2 | 31.7 | 78.0 |
>
> *Table A3: Comparison across different pretext tasks pre-train on K400-50k subset and fine tuned on UCF101.*
>
> -  Method Categories: We came up with two major taxonomies for design choices for context-based approaches (survey 50+ works), namely: 1) Data transformation during the pre-training stage: Spatial, Temporal, and Spatio-temporal, and, 2) Contrastive and Non-contrastive approaches.
>
> - c.1) Choice of methods: (i) Most of the approaches in SSL (40+ works) are built or shown on C3D, R3D & R21D only & have been widely used in the SSL action recognition. That's why the major focus of our study is on R21D which has the best performance amongst these three. (ii) Transformers: In the appendix, we show an in-depth analysis of VideoSwin which shows similar behavior that performance saturates even with training for longer duration. It shows generalization to other classes of models.
>
> - c.2) Choice of ablations: We agree that for different aspects different types of analysis can be more helpful. However, in our work, we focus on these four crucial tasks for the following reasons:
>
>   - Dataset size: plays a crucial role in the performance of downstream tasks in SSL from literature in this domain. That’s why it is important to study if the increase in the size of the pretraining dataset will proportionally reciprocate in performance improvement. Also, a general trend is to train models for a very long duration at the pre-training stage. We investigate if the longer duration impacts the gain in performance. We look across different stages of training for multiple architectures and across different pretext tasks.
>
>   - Task complexity: Existing works show that increasing complexity leads to better representation learning, and if the complexity is decreased, the network will optimize to suboptimal solutions. We analyze this aspect in more detail with several tasks and different models and provide some interesting conclusions.
>
>   - Robustness of SSL tasks: is important to analyze which pretext tasks is the more robust when it comes to real-world data that contains noise. We analyze which factors play a key role in the robustness of these methods against such domain shifts.
>
>   - Feature analysis: Complementary features are an important analysis to identify how more information can be extracted from models trained on SSL tasks in different ways. There’s no existing study that performs knowledge distillation on four different aspects and shows conclusive evidence that indeed complementary information can be extracted.
>
> - c.3) Choice of datasets: Action recognition datasets are mainly divided into two ways: 1) Appearance and 2) Temporal based. We selected one large-scale dataset from each for pre-training. For downstream we cover three datasets, two from appearance and one from temporal.  It covers all possible variations in both pre-training and downstream.
>
> - d) We will revise the literature study in related work such that it will be beneficial for the community.
>
>
> **W.3, 4 A table to summarize the latest studies on the same topic and their contributions and identify gaps. The authors also need to position their observations with respect to other studies on SSL and their findings.**
>
> We compare our work to a recent study [1] which attempts to take a step in this direction, but it is mainly focused on downstream learning, without exploring the self-supervision aspect which is one of the main goals in our study. The analysis is limited to one architecture.

---

> > ### Author Rebuttal · Authors · 2024-08-16
> >
> > **W.3, 4 Response Continued...**
> >
> > Against this, our benchmark has three significant differences: 1) We perform an analysis on training at both pre-training and downstream stages. 2) We keep all hyperparameters similar in terms of dataset size and number of frames for both pre-training and downstream for a fair comparison which in [1] since all tasks are pre-trained in different ways is not a fair comparison. 3) We look into multiple architectures - both CNN and transformer-based.
> >
> > [1] Thoker, Fida Mohammad et al. “How Severe is Benchmark-Sensitivity in Video Self-Supervised Learning?” ArXiv abs/2203.14221 (2022): n. Pag. (ECCV 2022).
> >
> >
> > **W.5 The term "feature analysis" and its context are not well explained, even though it is frequently used from the beginning.**
> >
> > Feature analysis: is an analysis of the complementary nature of features to identify how more information can be extracted from models trained on SSL tasks in different ways. The approach used in our work ensembles the knowledge from multiple teachers' features via a gradient-based approach. This work has two benefits: 1) Instead of just averaging, it uses adaptive weighting to weigh the importance of individual teachers for better guidance. 2) Due to this advantage, it can use any number of teachers to guide the student (Fig. 3 - b main paper).
> >
> > There’s no existing study that performs knowledge distillation on four different aspects and shows conclusive evidence that indeed complementary information can be extracted.
> >
> >
> > **W.6 Choice of methods is questionable, leading to several basic limitations: Due to the choice of models, the studied effects are limited to those models only. How can these findings be generalized to various other classes of models? Justification for selection of models.**
> >
> > - Most of the approaches in SSL (40+ works) are built or shown on C3D, R3D & R21D only & have been widely used in the SSL action recognition. That's why the major focus of our study is on R21D which has the best performance amongst these three.
> >
> > - Transformers: In the appendix (Table-5 supple), we show an in-depth analysis of VideoSwin which shows similar behavior that performance saturates even with training for longer duration. It shows generalization to other classes of models.
> >
> > - Additionally for the rebuttal, we perform an experiment with pre-training on ViViT on K400-50k, which achieves 33.6% on UCF101.
> >
> > **W.7  Ablations are limited to noise only, while many other perturbations exist. This needs to be justified.**
> >
> > We agree with the reviewer that a lot of perturbation exists. The focus of our study is to evaluate the robustness of SSL models on real-world environmental noises. Thus, we choose four noises. We want to add that we not only look into different noises but also at different severity levels of noises. In our scenario, it was five. In Fig 9 supple, we show the performance of all six models at different levels of noise. We will add more types of noises and revise our manuscript.
> >
> > **W.8 Citations to previous studies on the same topic are missing. Many studies have addressed SSL for videos.**
> >
> > We will include the latest studies in the related works. There are a lot of works. We tried to cite a few survey papers and a lot of approaches, but we might have missed some. We will revise the manuscript and include those.
> >
> > **W.9 More details on the distinction between generative vs. discriminative or equivariance vs. invariant-based, hybrid multi-modality approaches need to be covered. The field is vast, and many design choices need to be explained. It is advisable to cover a niche thoroughly, especially if it overlaps with futuristic trends, as this will be more helpful for the community.**
> >
> > We can include a section comparison between generative vs. discriminative or equivariance vs. invariant-based and multiple modalities approaches in related works. We do want to stress that the primary focus of our work is on SSL tasks for single-modality approaches.
> >
> > **W. 12 The use of methods across different experiments is not uniform; some are missing.**
> >
> > In task complexity analysis scenarios, it was difficult to judge the increase in complexity for contrastive works because it is mostly dependent on hyperparameters such as the number of negatives and positives in a batch. However, for non-contrastive tasks, the task complexity is properly defined. Thus, only non-contrastive approaches were shown for this study. However, in most analyses, we have included almost all the works. We selected some best works for a few experiments on preliminary analysis to restrict the number of experiments.
> >
> > **W. 14 In the abstract, it is claimed that "...we focus on six different methods along with six different network architectures...", but only 2 methods, ShuffleNet and R2+1D, are used.**
> >
> > In Tables 1 and 5 supple we show the results of six different models for six different architectures. We will revise the manuscript and bring the table in the main paper.
> >
> > **W. 15 Contradiction in use of large capacity networks in abstract vs paper.: in the abstract, it is stated, "...Furthermore, we extend these findings to Video Foundation models (ViFMs)...", but elsewhere it is mentioned, "...We do not include large capacity networks in our main benchmark in the interest of computational efficiency...".**
> >
> > In Table 6 (main paper), for comparing ViFMs, we did not perform any finetuning or linear probing; we directly used the models in a zero-shot manner. For knowledge distillation from ViFMs, we use the zero-shot logits to distill an R21D network (Table 7 main paper).
> >
> > **W. 10, 11, 13, 16 Improvement in Presentation and writing.**
> >
> > We will revise the manuscripts and improve the presentation of abbreviations, expand on captions of tables and figures, and grammatical errors in writing.

---

> > > ### Comment · Area_Chair_nDUZ · 2024-08-29
> > > **Reminder to response to author rebuttal**
> > >
> > > Dear Reviewer,
> > >
> > > The ddl for author and reviewer discussion is approaching. Please check the author rebuttal and leave some comments to respond to author rebuttal.
> > >
> > > Thanks,
> > >
> > > Your AC

---

> ### Author Response · Authors · 2024-08-23
> **Review Clarification**
>
> Dear Reviewer Kssv,
>
> We are sincerely thankful for the time and work you put into reviewing our paper. We hope our answers clarify your queries and if you have any more queries regarding the paper feel free to ask us any time. We will be glad to answer them.
>
> Sincerely,
> Authors of Paper 2028

---

> ### Comment · Reviewer_Kssv · 2024-08-25
>
> I thank the authors for addressing to the comments. I agree that authors have put effort to perform many experiments and cover the problem from different aspects. However, I believe conclusions are not generalizable. Please see the comments below:
> - (a) There are many representatives from each class of SSL methods (such as MVL [1]), such as pBYOL which performs better than CVRL, however, this is not considered.
> - (b) More weight is put on rot-net like models that are based on equivariance idea which is known to perform suboptimally to the invariance based contrastive methods or the current VideoMAE/InternVideo/OmniVAE like generative methods. All representative methods have not been used under different ablations.
> - (c) Section 4.3, only explored for pretext task, not for generative or distillation or hybrid models [1]. These questions are biased to pretext task based models which are not the preferred choice in current works. To generalize these experiments to "effect of dataset distribution" may be misleading for the multiple families of SSL methods. Moreover the effect of dataset size or data type may also influence the obtained results, which is not explored. The word pretext task is generalized to both contrastive and non-contrastive methods, when the contrastive models do not involve the pretext tasks. In Fig 2, the models are called the task such as the spatio-temporal model (RSPnet) while the task should have implied the nature of task irrespective of models and multiple models of the same family may be tested for individual task.
> - (d) Similarly section 4.2, only deals with pretext based methods and restricts the use of word task to pretext only tasks, which again makes trivial set of experiments. Similarly section 4.4, only studies the role of noise for studying the robustness which is again narrowed approach to the kind of out of distributional changes that generally occur in videos such as resolution, camera motion, blur, affine turbulence etc.
> - (e) Instead of shuffle-backbone, S3D-G or R-50 may be used which is more standard. SSL methods scale non-linearly with increase in size of training data, it would be useful to test if these results hold with 200k on k400 dataset.
> - Overall, paper attempts to do a comprehensive study, however, different parts of the problems are handled as per suitability, which has increased the number of experiments but do not provide a conclusive evidence on many aspects of the problem that are studied.
>
> W.3: Please see few more studies [2,3,4]
>
> [1] Wang, R., Chen, D., Wu, Z., Chen, Y., Dai, X., Liu, M., ... & Jiang, Y. G. (2023). Masked video distillation: Rethinking masked feature modeling for self-supervised video representation learning. In Proceedings of the IEEE/CVF conference on computer vision and pattern recognition (pp. 6312-6322).
>
> [2] Thoker, F. M., Doughty, H., Bagad, P., & Snoek, C. G. (2022, October). How severe is benchmark-sensitivity in video self-supervised learning?. In European Conference on Computer Vision (pp. 632-652). Cham: Springer Nature Switzerland.
> [3] Deng, A., Yang, T., & Chen, C. (2023). A large-scale study of spatiotemporal representation learning with a new benchmark on action recognition. In Proceedings of the IEEE/CVF International Conference on Computer Vision (pp. 20519-20531).
> [4] Yuan, L., Gundavarapu, N. B., Zhao, L., Zhou, H., Cui, Y., Jiang, L., ... & Gong, B. (2023). Videoglue: Video general understanding evaluation of foundation models. arXiv preprint arXiv:2307.03166.
>
> I have changed my rating

---

> > ### Author Rebuttal · Authors · 2024-08-30
> >
> > Thank you for your timely response and for providing us with another opportunity to improve our work. We have addressed the questions and concerns raised by the reviewer.
> >
> >
> > **W.1 & W.2 Performance on other representative classes of SSL such as MVL[1]/VideoMAE/OmniVAE and pBYOL.**
> >
> > We do agree with the reviewer that generative is another set of SSL uprising approaches. However, our study focus is context-based approaches. PRP approach selected in our work involves feature reconstruction as a pretext task which is analogous to generative/predictive approaches.
> >
> >  For rebuttal, we ran an additional experiment on Video-MAE with R21D backbone for fair comparison against other approaches. The results are shown in the table below:
> >
> > | Network    | Rot| VCOP | PRP | CVRL | TDL | RSP |  Video-MAE|
> > |---------------|-------|--------|------|------|------|------|------|
> > | R21D       | 41.2 | 51.5 | 46.2 | 61.2 | 31.7 | 78.0 | 76.2 |
> >
> > We observe that Video MAE outperforms other approaches apart from RSPNet. OmniVAE and Masked Video Distillation (MVL) are based on VideoMAE. MVD employs a multi-teacher-student approach. pBYOL is an image-based self-supervised learning approach. We will include a section of discussion with generative works and the result with VideoMAE in the revised version of the manuscript.
> >
> >
> > **W.3 Section 4.3 Generalization on multiple families of SSL methods. Performance with different dataset sizes. Generalization of the word - pretext task.**
> >
> > - We agree with the reviewer that the generalization cannot be made to other families of SSL methods such as generative. However, the focus of our study is on context-based approaches. We can revise the paper title to "Benchmarking Contextual Self-supervised Learning". We are working to include works on generative to make the study more generalizable across multiple families of SSL approaches.
> >
> > - We are currently running the experiments with 10k and 30k dataset sizes. Since the training requires both pre-training and finetuning, it will take some days to finish. We will revise the manuscript with those numbers.
> >
> > - In Fig. 2, we observe a similar pattern for other models of the same pretext task. We show the best-performing model for better visualization. We will include the other models in the revised manuscript.
> >
> > **W.4 Section 4.2 only deals with pretext-based methods. Section 4.4 inclusion of more noises.**
> >
> > In Section 4.2, the complexity of contrastive approaches increases complexity, involving an increase in batch size to include more positive/negative samples. We generally see an increase in batch size, which improves performance. An increase in complexity in non-contrastive approaches is not trivial. We study this aspect. In newer works, these pretext tasks are mixed with contrastive approaches, thus, making it an important factor to ponder upon.
> >
> > We are currently generating more noise samples for the UCF101 dataset. Since it's a frame-by-frame operation, it will take some time. We will revise the manuscript with the inclusion of a performance comparison on more noises.
> >
> > **W.5 Use of S3D-G or R50 backbones. Performance on 200k subset.**
> >
> > We have incorporated the suggestion to include R-50 and S3D backbones in our video SSL benchmark. We have launched experiments on 10k, 30k, 50k, and 200k subsets. We will continuously update the results in the comments as training is done for the subsets.
> >
> > For the rebuttal, we ran an additional experiment on the best approaches, one from non-contrastive and one from contrastive with R21D backbone. There’s a subtle performance gain of 1-2% for both tasks, however, the gain from 10k to 50k is much more significant. That’s what we observed in our study that approx. 50k was the optimal size to achieve the nearest best performance to full dataset size.
> >
> > | Method    | 10k | 30k | 50k | 100k | 400k |
> > |---------------------|-------|--------|------|------|------|
> > | PRP (Non-contrastive)      | 17.5   | 42.7 | 46.2  |  47.0 | 47.4|
> > | RSPNet (Contrastive)        | 70.9  | 76.4  | 78.0  | 80.0 | 80.2|
> >
> > *Table A2: Performance comparison on scaling the dataset size.*
> >
> > **W.6 Comparison with [2,3,4].**
> >
> > - Comparing against [2] which attempts to take a step in this direction, but it is mainly focused on downstream learning, without exploring the self-supervision aspect which is one of the main goals in our study. The analysis is limited to one architecture. Against this, our benchmark has three significant differences: 1) We perform an analysis on training at both pre-training and downstream stages. 2) We keep all hyperparameters similar in terms of dataset size and number of frames for both pre-training and downstream for a fair comparison which in [2] since all tasks are pre-trained in different ways is not a fair comparison. 3) We look into multiple architectures - both CNN and transformer-based.
> >
> > - Comparing against [3] we have three significant differences - 1) [3] focuses on all appearance-based datasets whereas we focus on both appearance and temporal-based datasets. 2) The focus of [3] is more on the dataset point of view (data diversity) whereas our focus is on in-depth analysis of pretext tasks in self-supervised learning. 3) [3] utilizes only one SSL approach to generalize comments on all SSL pre-training whereas our work provides a comprehensive study on six different SSL context-based approaches.

---

> > > ### Author Rebuttal · Authors · 2024-08-30
> > >
> > > Response continued...
> > >
> > > - [4] does share a similar observation that training on videos is important for downstream datasets where temporal reasoning matters such as SSv2. Comparing against [4], we have three significant differences: 1) [4] focus is on adaptation methods of image FMs whereas our focus is on self-supervised learning on video model backbones. 2) [4] doesn't involve any pre-training and uses ad-hoc networks on top of frozen weights whereas we perform an analysis on training at both pre-training and downstream stages. 3) [4] doesn't utilize a specific backbone which makes the comparison unfair across approaches, whereas we compare against similar backbones for fair evaluation.
> > >
> > > [1] Wang, R., Chen, D., Wu, Z., Chen, Y., Dai, X., Liu, M., ... & Jiang, Y. G. (2023). Masked video distillation: Rethinking masked feature modeling for self-supervised video representation learning. In Proceedings of the IEEE/CVF conference on computer vision and pattern recognition (pp. 6312-6322).
> > >
> > > [2] Thoker, F. M., Doughty, H., Bagad, P., & Snoek, C. G. (2022, October). How severe is benchmark-sensitivity in video self-supervised learning?. In European Conference on Computer Vision (pp. 632-652). Cham: Springer Nature Switzerland. [3] Deng, A., Yang, T., & Chen, C. (2023). A large-scale study of spatiotemporal representation learning with a new benchmark on action recognition. In Proceedings of the IEEE/CVF International Conference on Computer Vision (pp. 20519-20531). [4] Yuan, L., Gundavarapu, N. B., Zhao, L., Zhou, H., Cui, Y., Jiang, L., ... & Gong, B. (2023). Videoglue: Video general understanding evaluation of foundation models. arXiv preprint arXiv:2307.03166.
> > >
> > > We hope our answers clarify your queries and if you have any more queries regarding the paper feel free to ask us any time. We will be glad to answer them.

---

> > > > ### Comment · Reviewer_Kssv · 2024-09-01
> > > > **Response to Rebuttal**
> > > >
> > > > Thanks to the authors for replying to all comments and attempting to incorporate the missing experiments, however, I still feel the work is incomplete.
> > > > W1 & W2.: Understandably, the paper is focusing on context based methods only. However, neither the title and abstract of the paper say that, nor is it state-of-the-art. So motivation and usefulness of exploring mostly out-phasing methods is not clear.
> > > > It is a good work and may be useful if completed, suggestion is to take time to carefully incorporate the missing experiments and analysis.
> > > > I have increased my rating.

---

### Official Review · Reviewer_n2Lu · 2024-07-24
**Benchmarking video SSL**

**Rating:** 6
**Confidence:** 3
**Correctness:** The paper appears to be generally cor…

**Review:**

Supporting the systematic evaluation of video SSL is important and timely, and so is taking stock of video SSL research so far in one place. The authors' effort goes someway towards meeting the declared objectives; albeit, with some omissions (see Opportunities For Improvement). I also spotted couple statements positioned as significant findings that are either common sense or not 100% justified (e.g., lines 294-296 and lines 274-280, respectively).

**Strengths:**

- good effort towards compiling relevant video SSL literature in one place and subjecting it to systematic testing
- good use of tools (CKA maps and distillation) to uncover insights

**Additional Feedback:**

The wording "Inference" (e.g., line 167) is not ideal as inference has obvious connotations in this community.

**Clarity:**

Language at times did come across convoluted. Some of the insights can be better phrased.

**Documentation:**

Could you release code for the benchmark?

**Limitations:**

No limitations were discussed.

**Opportunities For Improvement:**

- transformer modelling with proper spatiotemporal attention is absent from the proposed benchmark. To me, a video SSL benchmark should help us understand what would this inductive bias (O(S^2T^2), see Patrick et al. [1]) bring to the table. It is necessary for SOTA coverage.
- the effect of data scale beyond what the authors have scoped in the paper should be investigated, even as one isolated experiment for computational economy reasons; the full size of Kinetics400 can in theory be utilised.

[1] Patrick et al. "Keeping your eye on the ball: Trajectory attention in video transformers." NeurIPS 2021.

**Relation To Prior Work:**

Related work is discussed sufficiently in Sec. 2.

**Summary And Contributions:**

The paper provides a benchmark suite for video SSL. The suite evaluates and compares a few video SSL methods across multiple quantitive and qualitative performance perspectives. Several insights are derived followed by recommendations in order to inform future research directions.

---

> ### Author Rebuttal · Authors · 2024-08-16
>
> We sincerely thank the reviewer for the valuable feedback and analysis of the paper. We have addressed the questions and concerns raised by the reviewer.
>
> **W.1. Transformer modeling with proper spatiotemporal attention is absent from the proposed benchmark. How does this inductive bias [1] bring to the table?**
>
> We will include and discuss spatiotemporal attention in related work. However, the work reported is not self-supervised so we can't comment directly on the aspect of self-supervised. We do think that the inclusion of inductive bias leveraging multiple frames or motion trajectories of objects can help in the reduction of training data for self-supervised and boost the performance as well. It will help in inducing additional temporal information which is very crucial for self-supervised learning as shown in our work as well.
>
> Additionally, we want to add that the primary focus of our work is on SSL tasks with video backbones common across multiple works. [1] is based on ViT an image backbone. However, we will add a section discussing the impact of the induction of spatio-temporal attention in the SSL benchmarks.
>
> **W.2. Effect of data scale beyond in paper. Full size of K400.**
>
> For the rebuttal, we ran an additional experiment on the best approaches, one from non-contrastive and one from contrastive with R21D backbone. There’s a subtle performance gain of 1-2% for both tasks, however, the gain from 10k to 50k is much more significant. That’s what we observed in our study that approx. 50k was the optimal size to achieve the nearest best performance to the full dataset size.
>
> | Method    | 10k | 30k | 50k | 400k |
> |---------------------|-------|--------|------|------|
> | PRP (Non-contrastive)      | 17.5   | 42.7 | 46.2  |  47.4|
> | RSPNet (Contrastive)        | 70.9  | 76.4  | 78.0  | 80.2|
>
> *Table A2: Performance comparison on scaling the dataset size.*
>
> **W.3 Could you release code for the benchmark?**
>
> All code, models, and data will be publicly available.
>
> **W.4 Word inference is not ideal as it holds different meanings.**
>
> We will revise the manuscript and update it with conclusions, observations or findings accordingly.

---

> > ### Comment · Reviewer_n2Lu · 2024-08-18
> >
> > I would like to thank the authors for their comments.
> > I have read their response. However, I still believe that treating spatiotemporal attention in a video SSL benchmark is important for SOTA coverage. There are multiple works that do SSL transformer training (couple examples for contrastive transformer training below) and it should be possible to do the same for the aforementioned spatiotemporal backbone.
> > I am willing to increase my score if this is addressed.
> >
> > - A Recasens et al. Zorro: the masked multimodal transformer. arXiv:2301.09595. 2023
> > - A Fuller et al. CROMA: Remote sensing representations with contrastive radar-optical masked autoencoders. Neurips 2024

---

> > ### Author Rebuttal · Authors · 2024-08-23
> >
> > Thank you for your timely response and for providing us with another opportunity to improve our work. We have incorporated the suggestion to include spatiotemporal attention in our video SSL benchmark. We are working on the Patrick et al. [1] and have launched experiments on 30k, and 50k subsets. We will continuously update the results in the comments as training is done for the subsets. We are working on the other suggested two models as well, and will add them to the revised version.
> >
> > [1] Patrick et al. "Keeping your eye on the ball: Trajectory attention in video transformers." NeurIPS 2021.

---

> > ### Author Rebuttal · Authors · 2024-08-28
> >
> > Thanks for providing us with another opportunity to improve our work. We share the results for suggested work on 30k and 50k subsets in Table *R1*. As Motionformer utilizes an image backbone, its performance is synonymous with the original paper underperforming than ViViT.
> >
> > | Network    | ShuffleNet| R21D | ViViT | Motionformer [1] |
> > |---------------|-------|--------|------|------|
> > | 30k | 68.2 | 76.6 | 31.5| 29.4 |
> > | 50k | 69.4 | 77.4 | 33.6 |31.2 |
> >
> > *Table R1: Performance comparison of spatio-temporal attention models.*
> >
> > We looked into the other two suggested works.  [1] we looked into the work and tried to run the repository however the codebase is incomplete.  Zorro [2] is a multimodal approach. Adapting to our work for fair comparison we looked only into the visual backbone of the work which is based on VideoSwin. We share the results for VideoSwin comparing against ShuffleNet and R21D on 30k and 50k subset datasets in Table *R2*.
> >
> > | Network    | ShuffleNet| R21D | VideoSwin |
> > |---------------|-------|--------|------|
> > | 30k | 68.2 | 76.6 | 46.3 |
> > | 50k | 69.4 | 77.4 | 52.5 |
> >
> > *Table R2: Performance comparison of VideoSwin.*
> >
> > [1] A Fuller et al. CROMA: Remote sensing representations with contrastive radar-optical masked autoencoders. Neurips 2024
> >
> > [2] A Recasens et al. Zorro: the masked multimodal transformer. arXiv:2301.09595. 2023

---

> > > ### Comment · Reviewer_n2Lu · 2024-08-30
> > >
> > > Thanks for your latest experiments. Could you please elaborate on your SSL implementation of Motionformer (e.g., architecture used for pretraining, # of prototypes, etc.) and required compute at your end?
> > >
> > > The two other references I pointed out were meant to only give you ideas for training Motionformer contrastively.

---

> > ### Author Rebuttal · Authors · 2024-08-30
> >
> > Sure. We incorporated the base Motionformer (Mformer) model in original paper (Table 6-last 3rd row main paper) with RSPNet Self-supervised learning technique for fair comparison against other backbones. We incorporated all the hyperparams specific to the model from the original repo confuration file (yaml). We chose RSPNet since it was the best performing model in our set of experiments.
> >
> > Transformer backbones doesn't work with 10k dataset size - which we similar behaviour with VideoSwin and ViViT as well. Thus, we trained the model on 30k and 50k dataset subset size. We trained the model on 2x32GB GPUs (Tesla V100-PCIE-32GB).
> >
> > We hope our answers clarify your queries and if you have any more queries regarding the paper feel free to ask us any time. We will be glad to answer them.

---

> > > ### Comment · Reviewer_n2Lu · 2024-09-01
> > >
> > > I'd be surprised if tuning Motionformer for contrastive learning would be as simple as borrowing components from RSPNet...
> > >
> > > As pointed out by other reviewers, this work is timely and promising but perhaps could use more attention (pun intended) to getting SOTA coverage right and up-to-date. I have, nonetheless, increased my score in line with effort the authors have made to improve this submission.

---

### Official Review · Reviewer_NhHv · 2024-07-24
**Review of submission #2028**

**Rating:** 5
**Confidence:** 4
**Correctness:** The paper is correct.

**Review:**

Overall the idea is interesting and needed for the video SSL community, but the experiments are not convincing (old models, old methods, small scale, easy benchmarks, not statistically significant).

Weaknesses:

- The analysis is based on methods, tasks, benchmarks and datasets that are rather old and small scale. The pretraining methods considered are based on contrastive learning and old pretext tasks, which are good for categorization-based downstream tasks but not for learning high quality temporal features. The future of SSL from video is oriented toward learning visual world models and therefore predictive models, and many recent baselines such as VideoMAE, SAE and V-JEPA in this category are missing in the study.
It would have been nice to have predictive methods incorporated, with one interesting and important axis of comparison being generative models against Joint-Embedding Predictive Architecture (JEPA) models.

- The set of tasks is restricted to action recognition and clip retrieval which are high-level categorization tasks, low-level tasks such as segmentation and tracking, and the ability to make long-range prediction, are very important components of video SSL models which cannot be evaluated with the current benchmark.

- The chosen datasets (HMDB51, UCF101) are rather small scale and relatively easy, especially since fine-tuning is preferred, despite linear evaluation being much more informative about SSL features. This is discussed in Line 128, why not use it ? Moreover, one of the conclusions of the paper is that temporal-based methods do not perform as well as spatial-temporal methods but this is biased by benchmarks that favor this ?

- The video architectures ((R(2+1)D), ShuffleNet V1 2.0X) are rather old and small scale, this is OK for computational resources and because VideoSwin is included in Appendix, but what about ViTs ?

- Analysis of dataset size: 100k samples is still a very small scale for SSL. I understand that the authors are limited by computational resources, but still today the best SSL on images models are trained on more than 10 millions of images (DINOv2) and the same trend is expected in video SSL. It is rather hard to draw any conclusion on scaling if the maximum size is 100k samples. Also, results from Table 1 tell more about the methods rather than data scaling: it is hard to compare gains from data scaling given how methods perform differently at the same scale. Line 39 says: “self-supervised pre-training parameters are kept consistent 40 across methods for a fair comparison” How is that fair as each method benefits from different parameters ? Finally, I am not sure making generalization for all methods from a class of method (e.g contrastive) is possible with only one representative method for each class.

- Analysis of task complexity: the notion of complexity here feels very arbitrary. It is represented by parameters that are very specific to chosen methods and I don’t see how it would generalize to the class of “spatio-temporal” of “temporal” methods. Moreover, the results are not statistically significant so it is hard to make any conclusion from this experiment (Table 3). What is the variance across training runs and methods ?

- Analysis of dataset distribution: again, here it feels more like comparing methods than class of methods. RSPNet is better than CVLR, can we conclude that spatial methods are better than spatial-temporal methods ? Also, the UCF dataset used is very toyish. The conclusion made is interesting (L207) but expected in theory and not supported by enough evidence in practice.

- Analysis of Robustness: Contrastive methods experience a bigger performance drop but they start from a better performance which makes it harder to interpret the results. Maybe this is an issue with the presentation, and absolute performance should be presented. The conclusion that contrastive approaches are less robust to noise is still interesting, what is your intuition for that ?

- Analysis of features: The knowledge distillation results are cool results but how is it useful and how to use it as a benchmark ?

- The final results after taking all observations into account are not convincing. The UCF101 dataset is not interesting and the performance is not even that good on HMDB51.

- L275: “we demonstrate that beyond a certain amount of training data, additional data provides diminishing returns for SSL in terms of performance improvement” Maybe you just don’t have the right recipe for scaling ?

**Strengths:**

- Self-supervised learning from video (SSL) is still a niche topic and methods are not evaluated according to standard protocols regarding data and pretext tasks, making it very difficult to know which approaches are superior to others for scaling and future progress. This benchmark proposes to alleviate this shortcoming and is therefore well-motivated.

- The range of evaluation axes considered and the variety of the analysis are broad enough to cover interesting aspects of evaluation that are not necessarily covered by other video SSL benchmarks. In particular measuring performance depending on dataset size and data distribution shift are important for SSL and not well-measured by most of the literature.

- The difference is made between appearance-based tasks (Kinetics) and temporal-based tasks (SSv2), which is appreciated as most of the literature consider Kinetics as a good benchmark for evaluating video models.

**Additional Feedback:**

No

**Clarity:**

The paper is well written. One small detail: sometimes Figures use T, S and ST, sometimes they use the corresponding method, which makes it hard to follow what is evaluated in practice.

**Documentation:**

There are enough details to reproduce the experiments.

**Limitations:**

Limitations are addressed but in the supplementary, I think it would be better to discuss them in the main paper.

**Opportunities For Improvement:**

Improve the quality and significance of the experiments, see main review.

**Relation To Prior Work:**

Relation to prior work is discussed.

**Summary And Contributions:**

This paper presents a new benchmark for evaluating self-supervised video representation learning approaches through several axes including pretraining dataset size, pretext task complexity, robustness to distribution shift and frames visual degradation. Several classes of methods are represented by existing methods and are evaluated on the different axes, allowing to draw conclusions about which class performs better for each of the axes.

---

> ### Author Rebuttal · Authors · 2024-08-17
>
> We sincerely thank the reviewer for the valuable feedback and analysis of the paper. We have addressed the questions and concerns raised by the reviewer.
>
> **W.1 Analysis is on old methods and on a small scale. Future of SSL from video is oriented toward learning visual worlds and predictive models. ( VideoMAE, SAE and V-JEPA) Comparison between one generative model against ours.**
>
> We do agree with the reviewer that generative/predictive is another set of SSL uprising approaches. However, our study focus is context-based approaches. PRP approach selected in our work involves feature reconstruction as a pretext task which is analogous to generative/predictive approaches.
>
>  For rebuttal, we ran an additional experiment on Video-MAE with R21D backbone for fair comparison against other approaches. The results are shown in the table below:
>
> | Network    | Rot| VCOP | PRP | CVRL | TDL | RSP |  Video-MAE|
> |---------------|-------|--------|------|------|------|------|------|
> | R21D       | 41.2 | 51.5 | 46.2 | 61.2 | 31.7 | 78.0 | 76.2 |
>
> We observe that Video MAE outperforms other approaches apart from RSPNet. V-JEPA is based on the VideoMAE idea of masking too. It incorporates a 3D multi-box masking and prediction. We will include a section of discussion with generative works and the result with VideoMAE in the revised version of the manuscript.
>
> **W.2 Only high-level categorization tasks are involved. Inclusion of low-level categorization tasks.**
>
> Our benchmark study focus is to have a more comprehensive and in-depth analysis of appearance and temporal-based datasets using video models. For low-level tasks, most of the approaches which are SOTA utilize image-based backbone because in general, they can encode frame-level features at a more fine-grained level which is a must for dense/low-level tasks. We will include a long-video understanding analysis in the revised version of our manuscript.
>
> **W.3 Selection of datasets. Why finetuning and not linear evaluation? Conclusion: Temporal-based methods do not perform as well as spatial-temporal methods but this is biased by benchmarks that favor this.**
>
> UCF101 and HMDB51 are the most common downstream datasets for most SSL works. We use finetuning instead of linear probing since finetuning outperforms linear probing by a good margin. Table-2 supple shows that finetuning outperforms Linear probing on both Shuffle and R21D across all tasks. Here we show it on R21D for K400-50K for Rot, VCOP and PRP tasks.
>
> | Network    | Rot| VCOP | PRP |
> |-------------|-------|--------|------|
> | Linear Probe       | 2.7 | 12.2 | 4.6 |
> | Finetun       | 41.2 | 51.5 | 46.2 |
>
> Temporal-based methods don't perform as well as spatio-temporal not because the benchmark favors them. It is because the temporal pretext tasks focus more on integrating spatial features across temporal dimensions that they give less weightage to individual frame fine-grained spatial information. Spatio-temporal pretext tasks incorporate both spatial and temporal dimension features. The task focuses on fine-grained spatial aspects of individual frames as well as temporally aggregated features. That’s why in general spatio-temporal outperforms temporal.
>
> **W.4 The video architectures ((R(2+1)D), ShuffleNet V1 2.0X) are rather old and small scale, this is OK for computational resources and because VideoSwin is included in Appendix, but what about ViTs?**
>
> - Most of the approaches in SSL (40+ works) are built or shown on C3D, R3D & R21D only & have been widely used in SSL action recognition. That's why the major focus of our study is on R21D which has the best performance amongst these three.
> - Transformers: In the appendix (Table-5 supple), we show an in-depth analysis of VideoSwin which shows similar behavior that performance saturates even with training for longer duration. It shows generalization to other classes of models.
> - Additionally, for the rebuttal, we perform an experiment with pre-training on ViViT on K400-50k, which achieves 33.6% on UCF101. ViT is an image model and our study focuses on video models.
>
>
> **W.5 Limitations of dataset size. How is that fair as each method benefits from different parameters ? Finally, I am not sure making generalization for all methods from a class of method (e.g contrastive) is possible with only one representative method for each class.**
>
> For the rebuttal, we ran an additional experiment on the best approaches, one from non-contrastive and one from contrastive with R21D backbone. There’s a subtle performance gain of 1-2% for both tasks, however, the gain from 10k to 50k is much more significant. That’s what we observed in our study that approx. 50k was the optimal size to achieve the nearest best performance to full dataset size.
>
> | Method    | 10k | 30k | 50k | 100k | 400k |
> |---------------------|-------|--------|------|------|------|
> | PRP (Non-contrastive)      | 17.5   | 42.7 | 46.2  |  47.0 | 47.4|
> | RSPNet (Contrastive)        | 70.9  | 76.4  | 78.0  | 80.0 | 80.2|
>
> We don’t keep hyperparameters very specific to the method. Only the spatio-temporal input to the model that is the video input size of using 16 frames and 112x112x3 frame size is kept similar. We can surely include more works but that would make the study infeasible. In terms of approaches, these approaches are the base approach on which several works have been built - temporal order prediction, rotation prediction, feature regeneration, and contrastive learning. We do agree that there are a lot of pretext tasks, however, the tasks selected in our work are the base works for a lot of recent works.

---

> > ### Author Rebuttal · Authors · 2024-08-17
> >
> > **W.6 Analysis of task complexity: Generalization to the class of “spatio-temporal” of “temporal” methods. Moreover, the results are not statistically significant so it is hard to make any conclusion from this experiment (Table 3).**
> >
> > Temporal tasks generally involve sampling at different levels or order predictions of video frames. VCOP utilizes reordering techniques. Pace Prediction an another approach that predicts the pace of video at different streaming rates.  Similarly for spatio-temporal tasks, the complexity is extended in both spatial and temporal dimensions.
> >
> > Table 3 provides an interesting insight into where the earlier trend was that the community kept increasing the task complexity until it ran out of GPU memory. Our analysis shows how tasks become unsolvable and start losing a spatio-temporal understanding of what action is happening after a certain point.
> >
> > **W.7 Conclusion from dataset distribution, can we conclude that spatial methods are better than spatial-temporal methods ? UCF dataset is very toyish. The conclusion made is interesting (L207) but expected in theory and not supported by enough evidence in practice.**
> >
> > Table 2 not only shows the redundancy of information in K400 that leads to a bare minimum gain of 0.5-1% gain. This helps in a huge reduction of computation, and furthermore, opens the future direction for intelligent data selection from pre-training datasets to have maximum gain in minimum time. Yes, we can conclude if the downstream dataset is appearance-based as well.
> >
> > In practice also, if the pre-training data aligns with downstream data, generally, the performance is better. For instance, performance on UCF101 will be lower if the model is pretrained on SSv2 against K400. On the other hand, for temporal datasets such as Diving48, SSv2 outperforms K400.
> >
> > **W.8 Absolute performance on analysis of Robustness. The conclusion that contrastive approaches are less robust to noise is still interesting, what is your intuition for that?**
> >
> > We include the table for absolute performance drop. We believe the relative drop is a better way to interpret since all models have different performances on clean and noisy datasets. The relative change in performance is a better metric than absolute for comparison.
> >
> > | Network    | Rot| VCOP | PRP | CVRL | TDL | RSP |
> > |---------------|-------|--------|------|------|------|------|
> > | Shuffle    | 11.9 | 29.2 | 16.9| 30.0 | 7.0 | 49.1 |
> > | R21D       | 36.8 | 41.7 | 13.8 | 13.2 | 23.3 | 24.3 |
> >
> > *Table A1: Absolute performance drop .*
> >
> > The intuition behind contrastive being less robust to noise comes from pre-training dependency which is dependent on multiple clean video samples in a batch for contrastive learning. Since non-contrastive tasks are less dependent on other clean video samples they show a better robustness than contrastive tasks.
> >
> > **W.9 Analysis of features: The knowledge distillation results are cool results but how is it useful and how to use it as a benchmark?**
> >
> > The idea to incorporate Knowledge distillation was based on its success on multiple tasks which was relatively unexplored. our study is the first to show that it successfully works along these four different axes. We believe this study will be valid with more and more diverse upcoming SSL tasks since it involves training on different datasets (appearance vs temporal) and training on different pre-training setups such as contrastive, non-contrastive, generative, different modality aspects, etc. Our study opens up future directions to explore more in-depth the aspect of knowledge distillation for the SSL domain.
> >
> > **W.10 UCF101 dataset is not interesting and the performance is not even that good on HMDB51. Right recipe for scaling?**
> >
> > Action Recognition - On UCF101 (Table 5-main paper), we outperform the previous approaches by a significant margin considering only 10% of the dataset. For HMDB (Table 5-main paper), The papers that are ahead of ours are AVTS, GDT in multi-modal and TCLR and CVRL in single modality. AVTS and GDT use two modalities, have more frames and AVTS also uses a bigger spatial size. CVRL uses a longer temporal sequence and bigger frame resolution compared to ours and TCLR utilizes 64 effective frames. Thus, effectively using only 16 frames our performance is quite competitive.
> >
> > We follow the recipe from [1] to autoscale a few parameters with an increase in dataset size. [1] have properly established the scaling rules.
> >
> > Goyal, Priya et al. “Accurate, Large Minibatch SGD: Training ImageNet in 1 Hour.” ArXiv abs/1706.02677 (2017): n. pag.
> >
> > **W.11 Limitations in the main paper.**
> >
> > We will revise the manuscript and include it in the main paper.

---

> > > ### Comment · Area_Chair_nDUZ · 2024-08-29
> > > **Reminder to response to author rebuttal**
> > >
> > > Dear Reviewer,
> > >
> > > The ddl for author and reviewer discussion is approaching. Please check the author rebuttal and leave some comments to respond to author rebuttal.
> > >
> > > Thanks,
> > >
> > > Your AC

---

> > > ### Comment · Reviewer_NhHv · 2024-09-02
> > > **Response to Rebuttal**
> > >
> > > I thank the authors for their clarifications and answering my questions. Some of my concerns have been addressed and I think to work is of good quality, however I am still not convinced by the significance of the setup:
> > >
> > > - About UFC101 and HMDB51, I maintain that these benchmarks are outdated, small scale and uninteresting when the goal is to make progress in SSL from video. For exemple the paper draws the conclusion that spatial-temporal methods are better than purely temporal methods, which is the case in their setup, but this might not be true at a much larger scale. Most of the findings/conclusions is the paper are only valid at small-scale, when the goal is only to perform action classification on the features, but these conclusion are not applicable to SSL from video in general.
> > >
> > > - About linear vs fine-tuning evaluation, the authors mention that they use fine-tuning because it works better, I think this is a wrong reason to use it. The goal of evals in SSL is to eval the quality of the learned representation, not to push for absolute performance, and linear eval provides a much better signal, and makes comparisons between methods much easier. The reason why linear was not used historically in video (while it was the most popular eval for SSL from images) is because SSL from video methods where too bad at the time.
> > >
> > > - About methods and backbones. I appreciate the effort to include VideoMAE results, however I still think that the focus on context-based methods, with old backbones, makes the impact of the proposed study very limited.
> > >
> > > I will increase my score to 5 for the effort the authors have made to improve the submission and answer my concerns.

---

> > ### Author Response · Authors · 2024-09-04
> >
> > Thank you for providing us with another opportunity to improve our work. We have addressed the questions and concerns raised by the reviewer.
> >
> > **W.1 Use of UCF101 and HMDB51 datasets. Whether conclusions will hold for bigger datasets.**
> >
> > UCF101 and HMDB51 are still a go-to choice for SSL action recognition performance evaluation. The performance on HMDB51 has still not reached 100%. The good performance on UCF is mostly due to it's spatial bias as shown in [1] and it helped us in disentangling the spatial and spatio-temporal aspects in self-supervised pre-training. Additionally, we have shown results with on Diving48 dataset as well (Section 4.3 main paper) for temporal action recognition dataset where time is more important than appearance. We have observations on pre-training on SSv2 dataset which is a rare explored direction but holds a lot of potential for temporal understanding of actions.
> >
> > The conclusions are drawn not only from quantitative point of view. We also added CKA maps to show qualitative reasoning as to why a certain observation holds. Those observations are purely based on pre-training which is at large-scale.
> >
> > **W.2 Linear vs Finetuning selection.**
> >
> > We agree with the reviewer that linear evaluation provides better signal for learned representations for downstream. Sorry for the confusion. We followed the similar protocol as previous work in SSL for downstream tasks. For action recognition, we followed finetuning and for clip retrieval we followed linear evaluation.
> >
> > **W.3 Performance with old backbones.**
> >
> > We covered all three major taxonomies - CNN (C3D, R3D & R21D), Transformers (Table -5 (supple) - VideoSwin & ViViT) and Video Foundation Models (Sec 5.3 (main paper) - ViFi-CLIP, X-CLIP, ViCLIP & LanguageBind) in our work. Since our work is focused on videos we employed models which are inherently 3D in nature. We based the selection of models on it's popularity in SSL works (40-50 works) & their wide usage in action recognition domain. We could include more models but that would make the study infeasible.
> >
> > [1] Choi, Jinwoo, et al. "Why can't i dance in the mall? learning to mitigate scene bias in action recognition." Advances in Neural Information Processing Systems 32 (2019).
> >
> > We hope our answers clarify your queries and if you have any more queries regarding the paper feel free to ask us any time. We will be glad to answer them.

---

> ### Author Response · Authors · 2024-08-23
> **Reiew Clarification**
>
> Dear Reviewer NhHv,
>
> We are sincerely thankful for the time and work you put into reviewing our paper. We hope our answers clarify your queries and if you have any more queries regarding the paper feel free to ask us any time. We will be glad to answer them.
>
> Sincerely,
> Authors of Paper 2028

---

### Official Review · Reviewer_CSgq · 2024-08-02
**This paper is a strong submission comparing SSL pre-text pre-training techniques for video models. Overall the paper would be a reasonable addition to the Datasets and Benchmarks track.**

**Rating:** 7
**Confidence:** 4

**Review:**

The paper addresses the challenge of comparing self-supervised learning methods for video representation due to the lack of a standard benchmark. It presents a multiple benchmarks for comparing critical aspects like dataset size, complexity, data distribution, noise robustness, and feature complementarity. The authors conduct extensive experiments using six methods, three network architectures, five datasets, and two downstream tasks. They offer insights and propose a method for UCF101 action recognition outperforming existing approaches with less training data.

In the past there have been extensive studies for pre-training video models for a particular end task - but there is a dearth of studies where recipes for pre-training models are compared for a variety of video tasks. The paper fills that gap by providing a comprehensive benchmark, with a diverse set of evaluations. By comparing multiple architectures and datasets, they provide generalizable findings for different video applications. I particularly liked the comparison of contrastive and non-contrastive pre-text techniques.

**Strengths:**

The paper has several significant contributions to the study of SSL for video representation. Here are some of its key strengths:
- **Comprehensive benchmark**: It provides a fair evaluation between various pre-text tasks for SSL for video. The evaluation is done over diverse settings (datasets, architectures, noise-levels)
- **Extensive experiments**: The number of experiments that a reader can draw conclusions from is large (especially given this is in the video domain). Plus, since most experiments aren’t evaluated on a particular downstream task, the conclusions drawn are probably more generalizable across a variety of video applications.
- **Insights and Findings**: The authors provide findings in every section: (1) If training time is a critical factor, contrastive tasks can be utilized to reduce pretraining time. However, these tasks might be less robust against data noise; (2) Ideally, a spatio-temporal pretext task should be used regardless of the data distribution. If this is not feasible, the pretext task should align with the nature of the pretraining dataset; (3) For models with limited capacity, increasing the pretraining dataset size and using complex pretext tasks offer no additional benefits; (4) For optimal performance, non-contrastive pretext tasks are preferable over contrastive ones; (5) Pretext tasks can learn complementary features across various model architectures, pretraining datasets, pretext tasks, and task complexities. This complementary knowledge can be distilled to obtain strong spatio-temporal features.
- **Clarity**: The paper is well structured, with detailed analysis in many places.

**Additional Feedback:**

N/A

**Clarity:**

Suggestions for improving clarity of the paper:
- It’s better to use the standard [1, 2, 3] for citations, instead of the current (1; 2; 3).
- Line 74: “… explores the domain of how much variation in …” is an incomplete sentence. Perhaps you can write “… explores the variation in …”
- Section numbering referring to supplemental doesn’t match the document (like in line 111, it should point to section 2.
- On line 185 can you please mention its figure 8 in supplemental (since there are other CKA maps too).
- Line 215: saying “In the main paper” makes it sound like this is not the main paper. Perhaps say “Here, we only discuss …”.
- The word “Inference” in ML is often used for model inference. I would replacing it with the word “Conclusions”. Possibly you could combine “Observations” and “Inference” into a single section called “Findings”, or something similar.
- What’s the reason for visually showing bar plots in a circle in Figure 3 (c). To me it looks more confusing when I want to compare student to teacher performance.
- Similarly Figure 3 (d) could be made into standard bar-char like Figure 3 (b). I am not sure whether adding a variety of ways to display bar-chart results adds anything.
- Figure 4 (c) / (d) are not referred to anywhere in the paper. I think it needs to go somewhere around line 261. Without it, that observations section seems like a qualitative analysis without any hard evidence.
- Table indicates that “Ours” is best performing. But it only is best performing for UCF101, and not for HMDB51. I think it would be better to highlight the best numbers in each column, and let the reader quantitatively decide for themselves which is the best performing method - and how far off is the performance of others against the winning approach. It is also good to highlight values within an error margin (for instance 97.3 and 97.0 are probably within an error margin).
- I think there should be an additional “Ours” row in Table 5 where the model is trained on full 200K+ K400 dataset. Currently it is unclear what is the source of low performance in HMDB51. A reader would always wonder whether it is because of a smaller training set size.
- Line 308: nit “frames is outperforms”

**Correctness:**

The benchmark/experiments seem to be sound. As discussed above, some motivation needs to be added why this set of experiments are chosen by the authors.

**Documentation:**

N/A

**Limitations:**

All limitations / shortcomings discussed in the "opportunities for improvement".

**Opportunities For Improvement:**

Overall the paper seems to be useful for researchers looking to pre-train models for video tasks. Although many useful tips and tricks are sprinkled around the paper - but some of the sections are not well motivated. It is not clear why the different axis measured for dataset size; pretext task complexity; noise are the most important ones for pre-training. One can argue that pre-training is done to create a common foundational model, and the most important aspect is to decrease overfitting - in which case noise and feature complementarity are more important than the dataset size and training time. I think the authors should consider adding reasons why these set of experiments were chosen.

Some other limitations / questions / confusions about the paper:
- The analysis in Table 3 shows results for both ShuffleNet/R21D. For a reader to make a discernment between the two architectures, it would be good to give computational complexity and memory footprint for both the models. R21D is better across the board, but are there cases where one might choose ShuffleNet. I am guessing there is a tradeoff of inference+training speed+memory footprint vs. accuracy.
- Where is the analysis mentioned on line 157: “b) performance across the different capacity of backbones”. The capacity analysis only happens in the next section (Sec. 4.2).
- It is unclear why knowledge distilled student’s performance peaks, and then drops off to a lower level as we increase the size of the dataset. (Figure 4 (a) / (b)). Line 233 mentions the fact, but it doesn’t offer any insight. Is it the capacity of the network that limits it consuming increasing amounts of complementary knowledge offered by the teachers? Is similar trend seen in training accuracy - or is this a case of overfitting?
- The Clip Retrieval and the Action Classification sections, as they are currently written, don’t seem to fit well together. The Clip Retrieval gives analysis for all four pretext tasks - but not sure what “Lessons (were) learned” from the previous section to arrive at this result. Although in the Action Classification section lessons about dataset size and orthogonality of features in distillation seems to have been applied to get results. Albeit with much lower performance than SOTA on K400.
- The paper suggests that the performance is plateauing with increasing dataset size (table 1). For many of the pretext tasks, it seems like it might following some log scale for performance against dataset size. It would be useful to know if that is the case - but this would need to be plotted with additional data points.
- Section 5.3 on “Extension of findings to Video Foundation Models (ViFMs)” seems to be lacking details. For instance I am not sure how UCF-101 classification is being done - is it by linear probing; or by fine-tuning? I am not sure if I can convincingly draw the conclusion that increasing the number of frames is the only source of success for ViFi-CLIP? Better ablations would need to be designed to draw that conclusion, i.e. keeping the architecture + epochs fixed while varying the number of input frames.
- I am not sure what conclusions to draw from the “complementary knowledge” Section 5.3. It’s a multi-teacher study where knowledge from two teachers are distilled in a student model. But there doesn’t seem to be a discussion on why certain pairings of teachers are better than others. If we take this study purely as an emperical finding, I’m not sure it would be valid even in a few months, given the pace of foundational models (SAM v2 is already out as I write this). Also, is “multi-student” on line 311 an error? - my understanding is that the student is always an R21D model.

**Relation To Prior Work:**

Yes. Section 2 details what is the difference between this paper and previous published work.

**Summary And Contributions:**

The paper provides a benchmark for comparing self-supervised learning techniques (pretext tasks) for video. They ablate different aspects of self-supervised pre-training of video models. This includes comparing performance with different pre-training: (1) dataset size; (2) task complexity; (3) temporal/spatial/spatio-temporal dataset distribution; and (4) noise. Additionally they perform knowledge-distillation based feature analysis to discover which pre-trained video models give complementary features. They also compare different pre-trained models on two downstream tasks

The paper explores 6 different kinds of pretext tasks from literature, spanning spatial, temporal, and spatio-temporal transformations. For each transformation, they explore one non-contrastive and one contrastive technique. The list of transformations included RotNet (classify video rotations); CVRL (contrastive learning on augmented clips); VCOP (classify video clip ordering); TDL (contrastive learning on triplets from same clip); PRL (discriminative-generator model for temporal up-sampling); RSPNet (spatio-temporal contrastive loss discriminating between videos and relative speed).

---

> ### Author Rebuttal · Authors · 2024-08-16
>
> We sincerely thank the reviewer for the valuable feedback and analysis of the paper. We have addressed the questions and concerns raised by the reviewer.
>
> **W.1. Reason for choosing dataset size; pretext task complexity; noise as most important ones for pre-training.**
>
> We agree with the reviewer that for different aspects different types of analysis can be more helpful. However, in our work we focus on these four crucial tasks for following reasons:
>
> - Dataset size: plays a crucial role in the performance of downstream tasks in SSL from literature in this domain. That’s why it is important to study if the increase in the size of the pretraining dataset will proportionally reciprocate in performance improvement. Also, a general trend is to train models for a very long duration at the pre-training stage. We investigate if the longer duration impacts the gain in performance. We look across different stages of training for multiple architectures and across different pretext tasks.
>
> - Task complexity: Existing works show that increasing complexity leads to better representation learning, and if the complexity is decreased, the network will optimize to suboptimal solutions. We analyze this aspect in more detail with several tasks and different models and provide some interesting conclusions.
>
> - Robustness of SSL tasks: is important to analyze which pretext tasks is the more robust when it comes to real-world data that contains noise. We analyze which factors play a key role in the robustness of these methods against such domain shifts.
>
> - Feature analysis: Complementary features are an important analysis to identify how more information can be extracted from models trained on SSL tasks in different ways. There’s no existing study that performs knowledge distillation on four different aspects and shows conclusive evidence that indeed complementary information can be extracted.
>
> **W.2. Computational complexity and memory footprint for both R21D and ShuffleNet. The tradeoff of inference+training speed+memory footprint vs. accuracy.**
>
> We add a table on comparison of flops and parameters and accuracy comparing between shufflenet and R21D on best non-contrastive - PRP and best contrastive - RSPNet. If memory footprint is a concern then definitely ShuffleNet can be utilized with only 2.5% of total GFLOPs against R21D and a 32% reduction in parameters.
>
> | Network    | Params | GFLOPs | PRP | RSPNet
> |---------------------|-------|--------|------|------|
> | PRP (Non-contrastive)      | 4.6M   | 1.1 | 41.1  | 68.8|
> | RSPNet (Contrastive)        | 14.4M  | 42.9  |58.9 | 78.0  |
>
> *Table A1: Params and GFLOPs comparison between Shuffle and R21D networks.*
>
> Furthermore, looking into the significance of smaller architectures, Table 1 in supple shows that smaller architectures could outperform bigger architectures such as C3D & R3D. It's pretext task-dependent. In addition to that, small architectures can be easily deployed in real-time settings as compared to bigger architectures. However, no study has been done on these architectures to the best of our knowledge.
>
> **W.3. Where is the analysis mentioned on line 157: “b) performance across the different capacity of backbones”.**
>
> Fig. 1 (in supple) shows the performance analysis across different capacities of backbones. We will revise the manuscript and include this in the main paper.
>
>
> **W.4. Clarity on why knowledge distilled student’s performance peaks, and then drops off to a lower level with an increase in dataset size. Is it the capacity of the network that limits it consuming increasing amounts of complementary knowledge offered by the teachers? Is a similar trend seen in training accuracy - or is this a case of overfitting?**
>
> Keeping the capacity of the network the same across four experiments with 10k, 30k, 50k, and 100k subset sizes, we observe the student performance is dependent on the teacher’s CKA maps. We observe that both teacher CKA maps are staggering grids then they learn complementary information and performance boosts. If either of the teachers has a dark multi-block pattern then it denotes that the particular teacher is not good enough to teach the student better and that’s why we see a decrease in performance. From Fig.3(a), it is evident from 10k and 100k teacher CKA maps that their patterns are more dark multi-block and CKA maps of 30k subset teachers are both staggering grids. In our set of experiments, we observe a peak at 30k. For other pre-training datasets, it could peak at different dataset subsets. The main point we wanted to convey is that complementary knowledge can be obtained.
>
> Relating to training accuracy, we see a boost in performance with an increase in dataset size shown in Table *A1*.
>
> | Method    | 10k | 30k | 50k |
> |---------------------|-------|--------|------|
> | PRP (Non-contrastive)      | 17.5   | 42.7 | 46.2  |
> | RSPNet (Contrastive)        | 70.9  | 76.4  | 78.0  |
>
> *Table A1: Performance comparison on scaling the dataset size.*
>
>
> **W.5. Lessons learned from Clip Retrieval section. Lower performance than SOTA on K400 on action recognition.**
>
> Lessons learned from clip retrieval -
>
> - Spatio-temporal tasks align best with downstream dataset features incorporating both spatial and temporal information for clip retrieval as against spatial and temporal only independent of pre-training data.
>
> - Contrastive learning performs better if both downstream and pre-training datasets are appearance-based. They are dataset-dependent when it comes to retrieving clips.
>
> - Temporal tasks aren't able to retrieve clips better when it is pre-trained on appearance-based datasets. It denotes that the task is not able to focus on spatial features which is important for clip retrieval.

---

> > ### Author Rebuttal · Authors · 2024-08-16
> >
> > **W.5. Response continued...**
> >
> > Action Recognition - On UCF101 (Table 5-main paper), we outperform the previous approaches by a significant margin considering only 10% of the dataset. For HMDB (Table 5-main paper), The papers that are ahead of ours are AVTS, GDT in multi-modal and TCLR and CVRL in single modality. AVTS and GDT use two modalities, have more frames and AVTS also uses a bigger spatial size. CVRL uses a longer temporal sequence and bigger frame resolution compared to ours and TCLR utilizes 64 effective frames. Thus, effectively using only 16 frames our performance is quite competitive.
> >
> > **W.6. Additional data points for subset size.**
> >
> > For the rebuttal, we ran an additional experiment on the best approaches, one from non-contrastive and one from contrastive with R21D backbone. There’s a subtle performance gain of 1-2% for both tasks, however, the gain from 10k to 50k is much more significant. That’s what we observed in our study that approx. 50k was the optimal size to achieve the nearest best performance to full dataset size.
> >
> > | Method    | 10k | 30k | 50k | 100k | 400k |
> > |---------------------|-------|--------|------|------|------|
> > | PRP (Non-contrastive)      | 17.5   | 42.7 | 46.2  |  47.0 | 47.4|
> > | RSPNet (Contrastive)        | 70.9  | 76.4  | 78.0  | 80.0 | 80.2|
> >
> > *Table A2: Performance comparison on scaling the dataset size.*
> >
> > **W.7. ViFMs - Inference on UCF-101 - is it by linear probing; or by fine-tuning? Is increasing the number of frames the only source of success for ViFi-CLIP?**
> >
> > In Table 6 (main paper), for comparing ViFMs, we did not perform any finetuning or linear probing; we directly used the models in a zero-shot manner. For knowledge distillation from ViFMs, we use the zero-shot logits to distill an R21D network (Table 7 main paper).
> >
> > From the ViFi-CLIP paper (Table 4), we observe that ViFi-CLIP outperforms approaches that are trained with 96 frames. The number of frames and the number of views also impact the performance. In terms of additional ablations, since our work focuses on computational efficiency, we used ViFMs as frozen networks to obtain zero-shot performance or logits for knowledge distillation tasks. ViFMs are huge networks that require a lot of computation memory (which was beyond the scope of our study), we will perform one ablation study comparing against other approaches.
> >
> >
> > **W.8.Conclusion on why certain pairings of teachers are better than others. Will this study be valid in a few months? Typo in Line 311 “multi-student”.**
> >
> > The idea to incorporate Knowledge distillation was based on its success on multiple tasks which was relatively unexplored. In our work, we attach CKA maps as to why certain pairings of teachers work, which qualitatively shows how good the learning of teacher models is. We believe this study will be valid with more and more diverse upcoming SSL tasks since it involves training on different datasets (appearance vs temporal) and training on different pre-training setups such as contrastive, non-contrastive, generative, different modality aspects, etc. We agree with the reviewer that more study is needed to explore the domain of the impact of knowledge distillation. However, our study is the first to show that it successfully works along these four different axes. Our study opens up the future directions to explore more in-depth the aspect of knowledge distillation for the SSL domain.
> >
> > Thanks for pointing this out. Multi-student is a typo. We will update it in the manuscript.
> >
> > **W.9. Suggestions for improvement on writing.**
> >
> > We will revise the manuscripts and improve the presentation of abbreviations, expand on captions of tables and figures, and grammatical errors in writing.

---

> > > ### Comment · Area_Chair_nDUZ · 2024-08-29
> > > **Reminder to response to author rebuttal**
> > >
> > > Dear Reviewer,
> > >
> > > The ddl for author and reviewer discussion is approaching. Please check the author rebuttal and leave some comments to respond to author rebuttal.
> > >
> > > Thanks,
> > >
> > > Your AC

---

> ### Author Response · Authors · 2024-08-23
> **Review Clarification**
>
> Dear Reviewer CSgq,
>
> We are sincerely thankful for the time and work you put into reviewing our paper. We hope our answers clarify your queries and if you have any more queries regarding the paper feel free to ask us any time. We will be glad to answer them.
>
> Sincerely,
> Authors of Paper 2028

---

### Author Rebuttal · Authors · 2024-08-17

We sincerely thank all the reviewers for the valuable feedback and analysis of the paper. The reviewers acknowledged the following strengths:

- **Comprehensive** benchmark study. (Reviewers: All)
- **Extensive** experiments across multiple axes and datasets. (Reviewers: CSgq, NhHv)
- **Good** efforts on literature compilation (Reviewers - n2Lu)
- Well-motivated and **good** insights. (Reviewers - CSgq, NhHv)

Here, we address the common major concerns raised by the reviewers.

**W.1. Reason for choosing dataset size; pretext task complexity; noise as most important ones for pre-training.**

We agree with the reviewer that for different aspects different types of analysis can be more helpful. However, in our work we focus on these four crucial tasks for following reasons:

- Dataset size: plays a crucial role in the performance of downstream tasks in SSL from literature in this domain. That’s why it is important to study if the increase in the size of the pretraining dataset will proportionally reciprocate in performance improvement. Also, a general trend is to train models for a very long duration at the pre-training stage. We investigate if the longer duration impacts the gain in performance. We look across different stages of training for multiple architectures and across different pretext tasks.

- Task complexity: Existing works show that increasing complexity leads to better representation learning, and if the complexity is decreased, the network will optimize to suboptimal solutions. We analyze this aspect in more detail with several tasks and different models and provide some interesting conclusions.

- Robustness of SSL tasks: is important to analyze which pretext tasks is the more robust when it comes to real-world data that contains noise. We analyze which factors play a key role in the robustness of these methods against such domain shifts.

- Feature analysis: Complementary features are an important analysis to identify how more information can be extracted from models trained on SSL tasks in different ways. There’s no existing study that performs knowledge distillation on four different aspects and shows conclusive evidence that indeed complementary information can be extracted.


**W.2 The video architectures ((R(2+1)D), ShuffleNet V1 2.0X) are rather old and small scale, this is OK for computational resources and because VideoSwin is included in Appendix, but what about ViTs?**

- Most of the approaches in SSL (40+ works) are built or shown on C3D, R3D & R21D only & have been widely used in SSL action recognition. That's why the major focus of our study is on R21D which has the best performance amongst these three.
- Transformers: In the appendix (Table-5 supple), we show an in-depth analysis of VideoSwin which shows similar behavior that performance saturates even with training for longer duration. It shows generalization to other classes of models.
- Additionally, for the rebuttal, we perform an experiment with pre-training on ViViT on K400-50k, which achieves 33.6% on UCF101. ViT is an image model and our study focuses on video models.

---

### Decision · Program_Chairs · 2024-09-26

**Decision:**

Reject

**Comment:**

This paper receives mixed reviews. While reviewers appreciate the importance of building a self-supervised representation learning benchmark for video tasks, some critical issues still remained for the current work, such as small evaluation dataset, only focus on context-based method, fine-tuning evaluation setting, limited evaluation backbones, missing evaluation on low-level tasks of tracking. The author provides a detailed rebuttal to these concerns. Some issues are well addressed, but some critical ones still remains: small evaluation dataset and unclear motivation of only focus on context based method. Based on these facts, the AC thinks this paper is not ready for publication before these critical issues are well addressed. Thus, the AC makes a reject recommendation to this paper.